# Diving into Self-Evolving Training for Multimodal Reasoning

## Abstract

Reasoning ability is essential for Large Multimodal Models (LMMs). In the absence of multimodal chain-of-thought annotated data, self-evolving training, where the model learns from its own outputs, has emerged as an effective and scalable approach for enhancing reasoning abilities. Despite its growing usage, a comprehensive understanding of self-evolving training, particularly in the context of multimodal reasoning, remains limited. In this paper, we delve into the intricacies of self-evolving training for multimodal reasoning, pinpointing three key factors: *Training Method*, *Reward Model*, and *Prompt Variation*. We systematically examine each factor and explore how various configurations affect the training's effectiveness. Our analysis leads to a set of best practices for each factor, aimed at optimizing multimodal reasoning. Furthermore, we explore the *Self-Evolution Dynamics* during training and the impact of automatic balancing mechanisms in boosting performance. After all the investigations, we present a final recipe for self-evolving training in multimodal reasoning, encapsulating these design choices into a framework we call M-STAR (**M**ultimodal **S**elf-evolving **Tra**ining for **R**easoning), built on MiniCPM-V 2.5. M-STAR achieves 59.5% accuracy on MathVista, surpassing the pre-evolved model by 6.9% absolutely without using additional human annotations. We believe this study fills a significant gap in the understanding of self-evolving training for multimodal reasoning and offers a robust framework for future research. Our policy and reward models, as well as the collected data, will be released to facilitate further investigation in multimodal reasoning.

## 1 Introduction

With the rapid advancement of Large Language Models, their reasoning abilities have improved significantly (Shao et al., 2024; Xin et al., 2024; Yang et al., 2024). This progress has been accompanied by a growing demand for more realistic and general reasoning capabilities. Multimodal reasoning, considered a fundamental skill in many real-world applications, such as intelligent agents (Liu et al., 2024c), robotics (Li et al., 2023; Liu et al., 2024b), and autonomous driving (Yang et al., 2023), exemplifies this trend. Multimodal reasoning requires Large Multimodal Models (LMMs) to understand various modalities beyond text. For example, visual mathematical reasoning (Lu et al., 2023) challenges models to analyze complex figures, diagrams, and charts, leveraging the provided information to perform reasoning tasks.

Despite these advances, the availability of human-annotated thought processes in multimodal scenarios remains limited, challenging the learning of multimodal reasoning. Consequently, self-evolving training, which utilizes model's own generation ability to iteratively tune and improve itself without external annotated data, has emerged as an appealing candidate to facilitate reasoning abilities. While research on self-evolving training has primarily focused on the text-only settings (Hosseini et al., 2024; Sun et al., 2024; Shao et al., 2024), its application in the multimodal domain, especially for reasoning tasks, has been limited with only a few sporadic examples (Fang et al., 2024; Dubey et al., 2024; Deng et al., 2024), and a unified framework has yet to be established.

We first identify the three key components of self-evolving training in multimodal reasoning, namely the **training method**, the use of **reward model** and the **prompt variation**, to build a clear and unified design space for searching the optimal strategies. Through massive controlled experiments, we find:

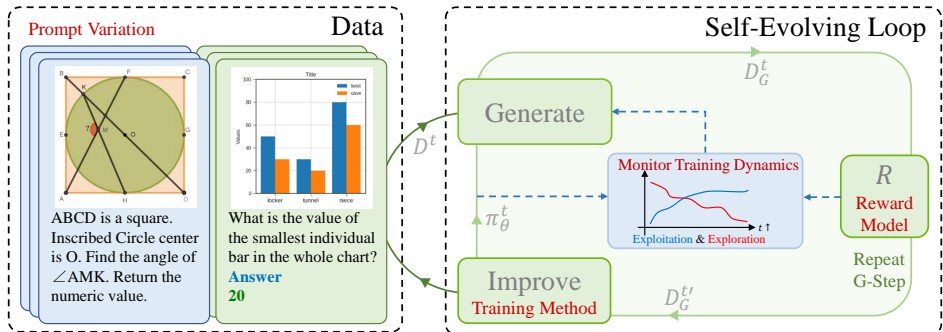

Figure 1: Overview of our self-evolving training framework for multimodal reasoning. We investigate the three essential design components of it, namely Training method, Reward model, and Prompt variation. Orthogonal to the static factors, the Dynamics of self-evoloution is also monitered, and provides control signals to the training process.

1. Optimizing the model from the last checkpoint is superior to retraining from scratch every time, and inheriting the optimizer states from the previous iteration also leads to performance improvements. Furthermore, each iteration should maintain an appropriate interval to traverse the queries in training set, neither too large nor too small (§3.2).

2. Including an extra reward model to re-rank and select the generated responses benefits a lot after filtering out incorrect responses, even if the reward model itself is not a qualified verifier (§3.3).

3. Adding more unlabeled queries helps only when having perfect reward signals (e.g., the oracle groundtruth answers), and it hurts the performance if the reward model does not generalize well on unseen data (§3.4).

Orthogonal to above static factors, we also dive into the dynamic evolving process of the model, namely the **dynamics of self-evolution** to see how model's behavior changes during the training. We find that although the model performance of greedy decoding increases, its exploration potential keeps decreasing through the training (§4).

Taking the self-evolving dynamics into account, we design a simple yet effective automatic mechanism, as shown in Figure 1, to dynamically adjust the temperature of sampling during training to balance the exploitation and exploration (§4.2). Experimental results show that this strategy, combined with the determined static design choices, effectively alleviate the loss of exploration throughout the training process and further boost the performance on both the in-domain and out-of-domain test sets.

## 2 OVERVIEW OF SELF-EVOLVING TRAINING FOR MULTIMODAL REASONING

Self-evolving training can be modeled as a general framework of reinforcement learning, where various algorithms can be formulated as a specific instantiation of RL, such as PPO (Schulman et al., 2017), STaR (Zelikman et al., 2022), ReST (Gulcehre et al., 2023) and ReST$^{EM}$ (Singh et al., 2023). Specifically, given a reward function $\mathcal{R}$, the objective of self-evolving training is to train the policy model $\pi_\theta$ to maximize expectation of reward $\mathcal{R}$:

$$\pi_\theta = \arg\max_{\pi_\theta} \sum_{i}^{L} \mathbb{E}_{x,o\sim\mathcal{D},\hat{y}_i\sim\pi_\theta[\cdot|x,o]}[\mathcal{R}(\hat{y}_i)], \tag{1}$$

where $x, o$ represent the query and image in the given training data $\mathcal{D}$, while $\hat{y}_i$ is a response sampled from the current policy model $\pi_\theta$. This standard RL objective, however, can be unstable to optimize and difficult to scale up, thus a popular algorithm adopted by recent works is to decouple the response rollout $\hat{y}_i \sim \pi_\theta[\cdot|x,o]$ and policy improvement into separate offline stages (Gulcehre et al., 2023; Singh et al., 2023): (1) *Generate*: the current policy model generates new responses $\hat{y}_i \sim \pi_\theta[\cdot|x,o]$; and (2) *Improve*: using the rewards to selects certain responses from the Generate step, which are then used to train the policy model with a standard supervised fine-tuning (SFT) loss. This way, the algorithm resembles Rejection Fine-Tuning (RFT, Yuan et al. (2023)) as it filters out negative

responses in a hard manner. Both steps are performed iteratively to strike a tradeoff between offline and online training. In many tasks such as mathematical problem-solving, there exists a unique, ground-truth answer $a^*$ which is utilized in the reward function, for example, Singh et al. (2023) directly adopts exact match to compute a binary reward by comparing $\hat{y}$ and $a^*$. In such an iterative training procedure, the objective at iteration $t$ is to obtain an improved policy model $\pi_\theta^{t+1}$:

$$\pi_\theta^{t+1} = \arg\max_{\pi_\theta^t} \sum_i^L \mathbb{E}_{x,o,y^* \sim \mathcal{D}, \hat{y}_i \sim \pi_\theta^t[\cdot|x,o]}[\mathcal{R}(a^*, \hat{y}_i)], \tag{2}$$

where the ground-truth answer input $a^*$ to the reward function $\mathcal{R}$ can be empty, for example, when dealing with unlabeled inputs, and then a reward model will be necessary to score $\hat{y}_i$.

**The Design Spaces**   There are different design choices to model and implement Eq. 2, for example, the design of reward function $\mathcal{R}$ and whether to incorporate additional unlabeled inputs without $a^*$ into training. Additionally, the training algorithms to perform this iterative process vary as well. For example, while Gulcehre et al. (2023); Xu et al. (2024b) initialize the model from the last checkpoint at each iteration, Zelikman et al. (2022); Singh et al. (2023) argue that initializing from the beginning checkpoint reduces overfitting and gives better performance empirically. Moreover, the iteration interval may matter as well – although the common practice is to process the entire dataset at every iteration and the performance saturates after few iterations, it may be suboptimal and a more online fashion with frequent iteration switch could potentially lead to improvements. Theoretically, a short iteration interval with inherited optimizer and learning rate scheduler from the last iteration will turn this iterative optimization into a standard online RL learning algorithm. Next, we investigate these three design spaces, *training method, reward model*, and *prompt variation*, aiming to summarize the best practices for each factor to faciliate multimodal reasoning learning.

## 3   DIVING INTO SELF-EVOLVING DESIGN COMPONENTS

In this section, we explore the three key components of self-evolving training, examining various strategies within each. We begin by outlining the general setup (§3.1 ), followed by a comprehensive analysis of each component to identify the best practices for multimodal self-evolution (§3.2-§3.4).

### 3.1   GENERAL SETUP

**Model:**   We base our exploration on MiniCPM-V-2.5 (Yao et al., 2024), a powerful, openly released LMM. MiniCPM-V-2.5 leverages LLaMA-3-8B (Meta, 2024) for its language model and SigLIP (Zhai et al., 2023) as its vision encoder, resulting in strong multimodal capabilities. Its performance on a wide range of multimodal benchmarks significantly surpasses previous openly released LMMs such as LLaVA (Liu et al., 2023; 2024a) and Qwen-VL (Bai et al., 2023). This superior performance makes MiniCPM-V-2.5 an ideal model for investigating self-evolving training in multimodal reasoning, with fewer risks of being constrained by the model's inherent capacities.

**Datasets:**   We utilize MathV360K (Shi et al., 2024), a high-quality and diverse multimodal reasoning dataset that includes 40K human-curated examples and 320K synthetic samples generated by GPT-4V as our seed training dataset. The images and queries in this dataset span various subjects in multimodal reasoning including algebra, arithmetic, geometry, logic, numeric commonsense and science. Specifically, we downsample 180K examples from MathV360K to serve as our labeled training set, while setting aside the remaining data as a unlabeled training set. For the labeled set, each data sample is composed of $(x, o, a)$ without the intermediate thought process annotation, while for the unlabeled set we only have access to $(x, o)$. This is a realistic setting as there are many multimodal SFT datasets with the final answer labels, but annotated thought processes are scarce. Our investigation will start with the labeled training set only, following existing practices (Singh et al., 2023; Zelikman et al., 2022), then we will study the impact of the unlabeled training data in §3.4.

**Warm-Up Phase to Unlock the Chain-of-Thought (CoT) Capability of LMMs:**   In our preliminary experiments, we found that open-source LMMs would directly output the answer given the query, while struggling to produce detailed chain-of-thought (CoT) reasoning processes. This may

originate from the the scarcity of high quality rationales in most existing multimodal SFT training datasets (Masry et al., 2022; Shi et al., 2024), which limits the ability of open-source LMMs to generate detailed, step-by-step reasoning. Self-evolving training, however, requires responses with varying intermediate steps to allow models to learn effectively from on-policy data. To address this issue, we initiate a warm-up phase as the first step before self-evolving training. Instead of prompting the model to answer questions directly, we prompt it to generate intermediate reasoning steps for a given triplet (question, image, and answer). For each triplet, we ask models to rollout 16 samples with temperature $= 1.0$. We then filter out results where the final answers do not match the ground truth and sample 100K from the generated dataset to create a warm-up CoT dataset $\mathcal{D}_w$ with correct answers. Finally, we fine-tune our models on this dataset, treating it as a standard RFT process. Our iterative self-evolving training process will then start from this model checkpoint after the warm-up training. For the prompt during the warm-up phase, please refer to Appendix A for more details.

**Training and Evaluation:** We adopt most training settings from Yao et al. (2024)(see Appendix B), using a constant learning rate of $1e - 6$ to train for 10K steps for all experiments. For all rollout phases in training, we sample 16 responses for each query and set the temperature of sampling as 1.0.

For evaluation, we employ two evaluation settings: an in-domain (ID) testset and an out-of-domain (OOD) one. For the in-domain test, we split 750 samples from the unlabeled part of MathV360K (Shi et al., 2024), using regular expression to extract and match the answers. For the OOD test, we assess our models using the testmini split of MathVista (Lu et al., 2023), a widely recognized benchmark for multimodal reasoning, using GPT-4 to extract and match the answers. We also keep an non-overlapping 250 samples from MathV360K as the global validation set in training.

## 3.2 TRAINING METHODS

As described in §2, there are multiple variants on how we would train to update the policy model. We decouple the variation dimensions by thinking of the gap between iterative training and online RL – when the iteration interval is small, the checkpoint at each iteration is initialized from one from the last iteration, and the optimizer as well as the learning rate scheduler is inherited between iterations, then iterative training becomes an online RL algorithm. Therefore, we cluster the variations by three factors: (1) **Model initialization**: when training is performed at the "Improve" step, the model can be initialized from either the last checkpoint (Xu et al., 2024b; Pang et al., 2024) or the beginning checkpoint before the first iteration (Zelikman et al., 2022; Singh et al., 2023); (2) **Iteration Interval**: while the common practice is to adopt a long iteration interval to process all the data queries for one iteration, we study the effect of having a shorter iteration interval, bringing it closer to online learning; and (3) **Continuous Optimization**: we propose to inherit the optimizers as well as the learning rate schedulers from the last iteration besides inheriting the model checkpoint, so that the optimization is continuous and closer to purely online learning algorithms. This way, we only have a global optimizer and learning rate scheduler essentially across the entire iterative training process. We note that while the variation on model initialization has been studied before, the other two factors, iteration interval and continuous optimization, have been rarely discussed in previous implementations of iterative self-evolving training, and they turn out to be important empirically as we will show next.

**Setup** We perform controlled experiments to study the effect of different training methods, thus in this experiment we use the labeled dataset only and simply adopt the binary exact-match reward between ground-truth answer $a^*$ and the generated answer. We compare with the most common iterative self-evolving algorithms ReST$^{EM}$ (Singh et al., 2023) and iterative RFT, which are specific instantiations of our training methods design space. To distinguish from the baselines, the variants with continuous optimization are named as *Continuous Self-Evolving*. To study the effect of iteration interval, we experiment with different percentage of all the queries per iteration, varying from [6.25%, 12.5%, 25%, 50%, 100%].

**Results** Table 1 presents the experimental results of various training methods. Overall, initializing training from the last policy model checkpoint $\pi_\theta^t$ and maintaining a continuous optimization process contribute the most significantly to the effectiveness of self-evolving training, particularly on MathVista. Continuous self-evolving achieves best performance both on the in-domain MathV360K test with 43.1% and on the OOD test set, MathVista, with 57.2%. We also see the importance of maintaining a proper interval to traverse the data queries. With a large interval, the training method

Table 1: Accuracy results (%) of self-evolving training using various training methods and iteration intervals. Iteration Interval (#) stands for the ratio of data we traverse in one iteration, and we also record the number of corresponding queries. $\mathcal{M}$ represents the policy model from which training is initialized in each iteration. $\mathcal{O}$ denotes whether the optimization process is continuous, i.e., the optimizer states and lr scheduler are inherited from the last checkpoint.

| Method | $\mathcal{M}$ | $\mathcal{O}$ | Iteration Interval (#) | MathV360K | MathVista |
|---|---|---|---|---|---|
| MiniCPM-V-2.5 | - | - | - | 13.6 | 52.4 |
| +warmup | - | - | - | 38.8 | 52.6 |
| SFT | - | - | - | 44.3 | 54.8 |
| Iterative RFT | $\pi_\theta^t$ | $\times$ | 100%(180K) | 42.3 | 55.7 |
| Rest$^{EM}$ | $\pi_\theta^0$ | $\times$ | 100%(180K) | 42.3 | 55.1 |
| Continous Self-Evolving | $\pi_\theta^t$ | $\checkmark$ | 100%(180K) | 42.2 | 56.7 |
| | | | 50%(90K) | **43.1** | 56.2 |
| | | | 25%(45K) | **43.1** | **57.2** |
| | | | 12.5%(22K) | 42.3 | 56.1 |
| | | | 6.25%(11K) | 41.0 | 56.8 |

becomes more closer to an offline one, and the model cannot get in-time update on data matching its current output distribution. On the other hand, switching over the *Improve* and *Generate* steps too frequently makes the learning unstable, leading to a lower score especially on the in-domain test set.

## 3.3 REWARD MODELS

In self-evolving training, the most common approach to reward function design uses a binary reward $\mathcal{R}(\hat{y}_i) = \mathbb{1}(\hat{a}_i = a^*)$, where $\hat{a}_i$ is the predicted answer inside $\hat{y}_i$ and incorrect responses are filtered out to maximize rewards. While effective, this sparse binary reward has limitations. It overlooks the quality of the intermediate reasoning steps within a response. Additionally, reward models trained from equal or higher capacity models than the policy model (Fried et al., 2022; Wang et al., 2024; Sun et al., 2024) can provide richer signals to improve the policy model's learning.

In this section, we introduce a Process Reward Model (PRM) (Lightman et al., 2023; Wang et al., 2024) for multimodal reasoning—the first of its kind, to our knowledge—and explore how integrating PRM can enhance reward design and whether it can improve policy model learning in self-evolving training for multimodal reasoning. To incorporate the reward scores into the objective of self-evolving training, the reward function is reformulated as:

$$\mathcal{R}(\hat{y}_i) = \mathcal{H}(\mathbb{1}(a^* = \hat{a}_i) \times \mathcal{R}_p(\hat{y}_i)) \tag{3}$$

$$\mathcal{R}_p(\hat{y}_i) = \min(f(s_i^0), f(s_i^1), ..., f(s_i^m)) \tag{4}$$

Here, $\mathcal{H}$ is an operation that processes responses based on the final reward scores, where we ensure all responses are correct by matching the ground truths, and $\mathcal{R}_p(\hat{y}_i)$ represents the process reward score for each sampled response. The function $f(s_i^k)$ denotes the reward score at each intermediate step. Following Lightman et al. (2023), we use the $\min$ operation to aggregate stepwise rewards.

**Setup** We conduct controlled experiments to assess the impact of incorporating the Process Reward Model (PRM) into self-evolving training and explore how best to utilize the reward signals provided by PRM. Notably, before applying PRM, responses are pre-filtered based on their final answers to ensure consistency and quality during training. To train our PRM, we use Monte Carlo rollouts starting from prefixes with partial reasoning steps (Wang et al., 2024) to generate the training data. Specifically, we sample 16 responses per question and complete each step 8 times to obtain step-level annotations. For additional details on the training process of our PRM, please refer to Appendix C. We evaluate two different $\mathcal{H}$ operations: (1) Top-K: Pick the top-K correct responses according to their reward scores, and (2) Filtering by a Threshold $\alpha$: Filtering out sampled responses with lower aggregated rewards than $\alpha$. The optimal value of $\alpha$ is 0.2 which is determined by enumerating it with an interval of 0.1 on the validation set. Additionally, we investigate how varying the value of K in Top-K affects training, as it represents a trade-off between the quality and diversity of the samples. According to §3.2, we fix training methods as continuous self-evolving with 45k interval and set continuous self-evolving, with or without randomly selected correct responses as our baselines.

Table 2: The results of self-evolving training with PRM and different strategies to leverage reward scores. $\mathcal{H}$ stands for the method to further pick out high-quality responses from the correct rollouts: (1) Top-k is we select K correct responses with highest rewards, and (2) $> \alpha$ is we pick out the correct responses with reward scores larger than $\alpha$.

| Method | $\mathcal{H}$ | PRM | MathV360K | MathVista |
|---|---|---|---|---|
| Continuous Self-Evolving | - | $\times$ | 43.1 | 57.2 |
| + Random Selection | Random-2 | $\times$ | 41.0 | 55.5 |
| +PRM-based Selection | $> \alpha$ | | 43.8 | 57.5 |
| | Top-1 | | 43.0 | 59.0 |
| | Top-2 | $\checkmark$ | **45.3** | **59.2** |
| | Top-4 | | 44.0 | 58.4 |

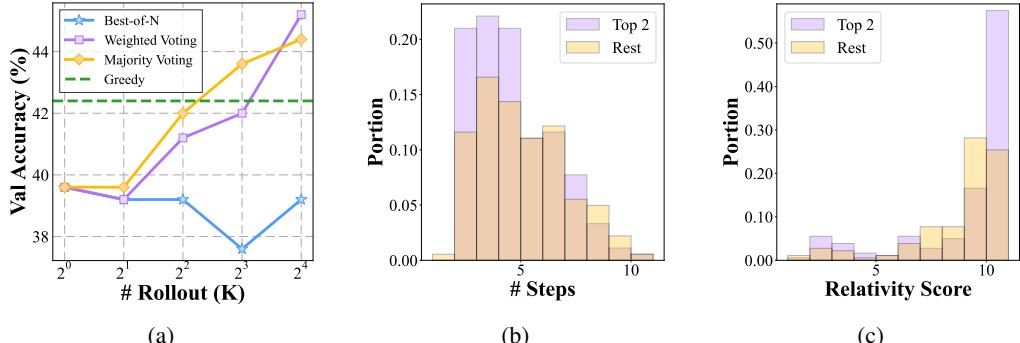

| (a) | (b) | (c) |

Figure 2: **(a)**: Accuracy on the val. set of greedy decoding and three selection strategy across different numbers of rollouts; **(b)/(c)**: Distribution of average # of steps and average relativity score annotated by GPT4-o of Top 2 and the rest responses re-ranked by rewards, we only calculate on correct ones.

**Results** Table 2 presents the results of integrating the PRM into self-evolving training, along with the impact of different $\mathcal{H}$ choices. Continuous Self-Evolving with PRM using Top-2 achieves the best performance in both the ID and OOD tests, with scores of 45.3% and 59.2%, respectively. Compared to training without PRM, most instances of self-evolving training with PRM show improved performance, especially in the OOD test. Interestingly, randomly selecting a subset of correct responses actually leads to worse performance than continuous self-evolving, suggesting that even correct answers can be noisy. Random selection may increase the proportion of these noisy samples, undermining the effectiveness of self-evolving training.

In terms of leveraging PRM, we found that using Top-K with a moderate number of responses—a re-ranking operation to select K correct responses with the highest-quality intermediate steps—outperforms filtering by a threshold. The results also highlight the importance of balancing the quality and diversity of sampled responses in self-evolving training. Selecting K = 2 strikes this balance well, ensuring both response diversity and high-quality reasoning steps for each question.

**What makes PRM work for self-evolving training?** To pursue deeper insights into the role of PRM in self-evolving training, we conduct an analysis presented in Figure 2. Based on the experimental results from §3.3, we explore PRM's impact from two key perspectives: (1) Can PRM help the model to select out correct responses among different numbers of rollouts? (2) How different are the Top 2 and the rest correct solutions re-ranked by reward scores? We use the first checkpoint after warmup $\pi_\theta^0$ as policy model to sample 16 responses for each question in the validation set with temperature=1.0 and reveal the behaviors of PRM in these samples.

We evaluate the verification ability of our PRM using two metrics, Best-of-N (BoN) and weighted voting (Sun et al., 2024), which are commonly employed to assess the performance of reward models. Surprisingly, as shown in Figure 2a, our PRM underperforms in both metrics. Notably, BoN and weighted voting yield worse results than vanilla majority voting when $N < 16$. We speculate that

this is due to the lack of high-quality step-level annotations compared to text-only reasoning tasks. These findings suggest that our PRM is not an effective **verifier**.

To understand why our PRM can still significantly contributes to self-evolving training despite its weaker verification abilities, we analyzed the distribution of other metrics for the top-2 selected responses compared to other correct responses. We approached this from two perspectives: the average number of reasoning steps, and how much a response is directly relevant to the question annotated by GPT-4o (see Appendix D), since we do not find incorrect steps but find some irrelevant steps after randomly checking some examples . As shown in Figures 2b and 2c, the responses re-ranked by our PRM generally have fewer reasoning steps and more relavant to the query. This highlights the **precision** of our PRM in recognizing genuinely high-quality responses. Therefore, our PRM acts as an effective **reranker**, precisely identifying top-quality responses. This precision is especially critical in self-evolving training, where responses are already filtered by ground-truth answers, and the ability to accurately assess the quality of reasoning steps becomes vital.

In addition to the aforementioned analysis, we also investigate why leveraging $\alpha$ to filter responses with lower reward scores performs worse than Top-K. The results indicate that, even with the optimal threshold value determined from the validation set, it tends to either retain or filter out all responses for each query, which reduces diversity and makes the learning process more challenging. This further supports the conclusion that **our PRM performs better as a Reranker than as a Verifier**.

### 3.4 PROMPT VARIATION

In this section, we explore how prompt variation affects self-evolving training. There are two primary types of prompts: labeled prompts and unlabeled prompts. Labeled prompts come with annotated ground truth answers, which can be used to filter out incorrect responses during training. In contrast, utilizing unlabeled prompts in self-evolving training is more challenging due to the absence of ground truth annotations. To maintain the quality of unlabeled prompts in training, surrogates like reward scores or pseudo labels must be employed. Meanwhile, unlike labeled prompts, unlabeled prompts are not be trained in SFT period, which increases the difficulty of learning for policy models.

**Skylines: Unlabeled Prompts with Oracle Reward Signals**    The coupling of these additional factors introduces complexity, making the effective use of unlabeled prompts less predictable. To dissect these factors, we start by establishing a baseline with "skyline" experiments, where both the unlabeled prompts and their ground truth answers are available but not used during the SFT phase. These unlabeled prompts with oracle reward signals serve as an intermediate difficulty between fully unlabeled and labeled prompts, providing insight into the challenges of training with unlabeled data.

**Unlabeled Prompts**    We incorporate unlabeled prompts into self-evolving training. To ensure the quality of sampled responses for these prompts, we use weighted voting to ensemble the predictions from different responses, treating the ensembled prediction as a pseudo label $\tilde{a}$. This pseudo label is then used to filter out responses with conflicting predictions, ensuring consistency. Following the best practices outlined in §3.3, we apply PRM as a reranker to select the top-2 responses among those with the predicted answer $\tilde{a}$. These unlabeled prompts are then mixed with labeled prompts for self-evolving training. Additionally, since learning from unlabeled prompts is more challenging for policy models, we investigate the optimal stage to introduce them into training to better understand their impact on model performance. We maintain a training interval of 45k prompts and adjust when unlabeled prompts are introduced into the self-evolving training process. Specifically, we introduce unlabeled prompts after [0%, 25%, 50%, 75%] of the total training process.

**A Glimpse at Unlabeled Prompts: Potential Efforts to Make Them Effective**    Table 3 presents the results of incorporating unlabeled prompts with and without oracle reward signals.

When training relies solely on oracle reward signals without integrating the PRM, continuous self-evolving with unlabeled prompts outperforms standard continuous self-evolving trained only on labeled prompts in the out-of-domain test but underperforms in the in-domain test. This indicates that additional prompts help the model generalize better to underrepresented questions but also increase the risk of forgetting previously learned information.

However, after combining with our PRM, all policy models perform worse than our best model trained exclusively on labeled prompts in both benchmarks, even when oracle reward signals are provided.

Based on the analysis in §3.3, this occurs since our PRM is unable to verify responses without ground-truth answers, and its generalization remains a concern.

When examining the timing for introducing unlabeled prompts, we find that adding them from the beginning helps mitigate the negative impact on model performance, compared to introducing them midway through the process. However, when unlabeled prompts are introduced later in the training process, they participate less in the overall training, leading to better results simply due to their limited involvement. This suggests that, without sufficient surrogate supervision (e.g., reward signals), introducing unlabeled prompts during the middle stages of self-evolving training can harm the process, potentially causing a deviation in the policy model's distribution.

Table 3: Results of involving unlabeled data. $T_{\texttt{mixin}}$ denotes when to mixin the unlabeled data. The use of PRM follows §3.3, except we first get a pesudo "ground truth" through weighted voting on unlabeled prompts.

| Oracle | PRM | $T_{\texttt{mixin}}$ | MathVista | MathV360K |
|---|---|---|---|---|
| - | × | - | 57.2 | 43.1 |
| - | ✓ | - | 59.2 | 45.3 |
| ✓ | × | 0% | 58.2 | 42.5 |
| ✓ | ✓ | 0% | 59.1 | 42.9 |
| × | ✓ | 0% | 58.2 | 43.3 |
| × | ✓ | 25% | 57.6 | 42.4 |
| × | ✓ | 50% | 58.2 | 42.9 |
| × | ✓ | 75% | 58.8 | 45.0 |

## 4 DYNAMICS OF SELF-EVOLUTION AND THE FINAL RECIPE

So far, we have explored the impact of three pivotal factors within our design space, leading to established best practices for learning multimodal reasoning – we adopt continuous self-evolving training coupled with a reward model to help data selection as described in §3.3, and we perform the training process on SFT datasets with final answer annotations. In this section, we delve even deeper into the current self-evolution strategy to better understand the bottlenecks. Instead of analyzing from a design space perspective as previously, we now fix the design parameters and focus exclusively on the training dynamics during the model's self-evolution. This shift in focus allows us to examine the process from an orthogonal angle, providing further insights into the underlying mechanisms that drive or impede progress in multimodal reasoning capabilities.

### 4.1 MONITORING THE TRAINING DYNAMICS

Intuitively, two critical conditions must be met for the success of self-evolving training: (1) the presence of high-quality candidate responses generated by the model, otherwise self-evolving will not work no matter how strong the reward is; and (2) the reward function's ability to effectively distinguish and prioritize these high-quality responses. These conditions align with the traditional reinforcement learning concepts of *exploration* and *exploitation*. Apparently, both exploration and exploitation capabilities are dynamic targets in self-evolving training, as the policy model evolves and the distribution of rollout responses changes with each iteration. To better understand these training dynamics, we propose tracking and visualizing four metrics:

- *Greedy Accuracy*: the model's accuracy with greedy decoding. We track this metric for reference to compare with other metrics.

- *Pass@K Accuracy*: the percentage of samples for which the model produces at least one correct response when sampling $K$ candidates. This metric measures the model's exploration ability.

- *(Pass@K - Greedy) Accuracy*: the difference between Pass@K and Greedy accuracy. Typically, Pass@K is an upper bound of Greedy Accuracy, and the gap roughly reflects the percentage of samples where the model, while failing in greedy decoding, can generate a correct response when sampling more candidates. This gap is crucial for the success of self-evolving training—a zero gap indicates that the model fails to explore correct responses for the current failure cases, suggesting that further training is unlikely to yield significant improvement.

- *Reward-Pass@2*: the percentage of samples for which there exist correct responses among the top 2 responses ranked by the reward model. This metric directly reflects the exploitation efficacy of the reward model for the current policy. We choose Pass@2 since our training strategy involves selecting the top 2 responses using the reward model (§3.3).

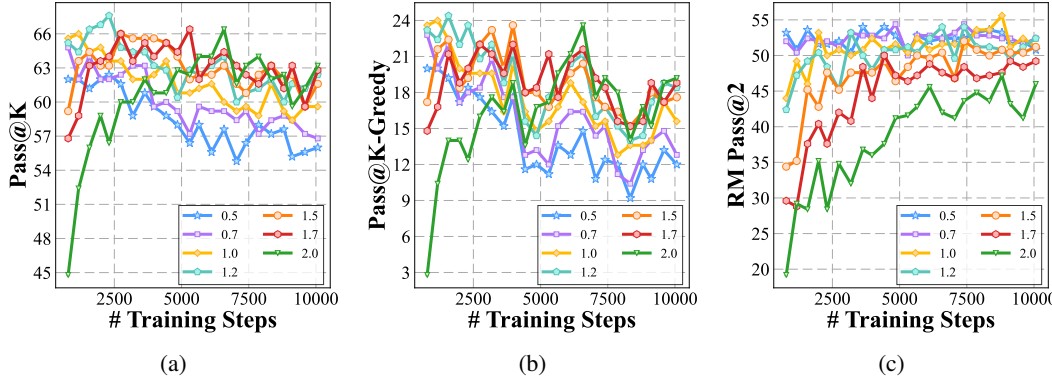

(a)  (b)  (c)

Figure 4: **(a)**: Pass@K decreases for all different temperatures; **(b)**: The gap between Pass@K and Greedy Decoding; **(c)**: The Reward-Pass@2 saturates quickly. All metrics, including the greedy decoding accuracy, are calculated on validation set.

Specifically, after each training iteration of our current optimal strategy, we sample 16 responses from the model checkpoint on the validation set, with the temperature range set to $t = [0.5, 0.7, 1.0, 1.2, 1.5, 1.7, 2.0]$. We analyze with varying temperatures as temperature is a key hyperparameter for the generation diversity and model's exploration.

**Results:** Figure 3 shows a clear trend where, as training progresses, the Pass@K metric continuously declines while greedy accuracy improves. This pattern indicates the loss of exploration ability, which hampers the model's potential for continuous improvement and may lead to performance saturation. These observations are consistent with findings in text-only settings as reported by Wu et al. (2024). In Figure 4a we analyze Pass@K accuracy at various temperatures and observe a significant trend: despite a general decay in exploration ability, larger temperatures tend to resist this decline more effectively, allowing the model to maintain a stronger ability to explore in the mid to late stages of training. This observation suggests that the optimal temperature for training may need to be dynamically adjusted throughout the self-evolving process, rather than being fixed at the outset as is currently common practice. In Figure 4b we plot the (Pass@K - Greedy)

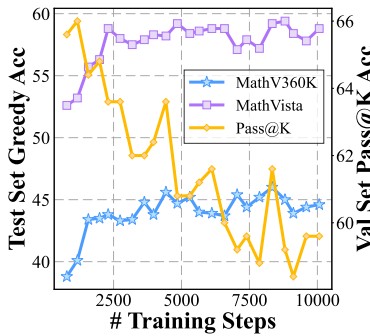

Figure 3: The opposite trend of Greedy Decoding Accuracy and Pass@K.

accuracy, this phenomenon becomes even more pronounced, indicating that the model's exploration during training is converging to greedy decoding. Additionally in Figure 4c, we observe that the Reward-Pass@2 metric initially increases but quickly reaches a plateau, indicating that the reward model's capacity to exploit further diminishes as training progresses. This limitation could be due to both the reduced exploration ability and the inherent constraints of the reward model. Next, we fix the reward model as a control variable and ask, **how can we enhance exploration to allow the reward model to exploit more effectively?**[1]

## 4.2 M-STAR– FINAL RECIPE WITH OPTIMAL DESIGN CHOICES & ADAPTIVE EXPLORATIONS

Reward-Pass@2 closely relates to the effectiveness of our self-evolving training strategy since our method selects top responses ranked by the reward model, and Reward-Pass@K directly reflects the quality of these 2 responses.[2] While Reward-Pass@2 naturally measures exploitation when the policy

---

[1]While improvements to the reward model could also enhance Reward-Pass@2, we reserve it for future work.

[2]We note that there is a slight mismatch between Reward-Pass@2 and our training strategy, as we pre-filter responses using the ground-truth answer before the reward model reranks them. Ideally, a more aligned metric would measure the CoT reasoning quality of the top 2 responses, both containing correct answers. Given that there is no reliable method to score the quality of the thought processes, we consider Reward-Pass@2 as a reasonable approximation which turns out to be effective empirically.

Table 4: Performance of M-STARcompared with baselines and methods considering only static components. We highlight the relative improvement of M-STAR over the pre-evolved model, i.e., the "+warmup" row.

| | MathV360K | MathVista |
|---|---|---|
| *Baselines* | | |
| MiniCPM-V-2.5 | 13.6 | 52.4 |
| + warmup | 38.8 | 52.6 |
| SFT | 44.3 | 54.8 |
| ReST$^{EM}$ | 42.3 | 55.1 |
| Iterative RFT | 42.3 | 55.7 |
| *Static components only* | | |
| Cont. optim. | 43.1 | 57.2 |
| + PRM Re-Rank | 45.3 | 59.2 |
| *Automatically tuning the temperature $T$* | | |
| M-STAR (*Pass@K*) | 42.8 | 58.0 |
| M-STAR (*Reward-Pass@2*) | 45.9 (+7.1) | 59.5 (+6.9) |

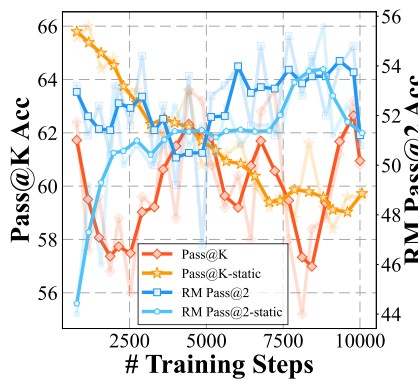

Figure 5: Comparing the Pass@K and Reward-Pass@2 metrics with the optimal static training progress, which fixes temperature $T = 1.0$. We use Savitzky-Golay filter (Savitzky & Golay, 1964) to smooth the curves.

is fixed, the absolute value of this metric actually encapsulates both exploration and exploitation – its value would be low if the model fails to explore high-quality candidates.

Therefore, we hypothesize that enhancing the Reward-Pass@K scores for the current iteration through varied configurations could potentially improve the efficacy of self-evolving training. We fix reward model as a control variable and focus on modifying the model's exploration capabilities to achieve this objective. Analysis in §4.1 suggests that the temperature, which is crucial for exploration, may require dynamic adjustment. Thus we propose to adjust the temperature automatically at each iteration based on the validation Reward-Pass@2 scores. This aims to optimize exploration so that the selected responses are of higher quality, potentially enhancing overall training effectiveness.

Specifically, we adjust the temperature per two iterations, and pick the temperature from 0.3 to 1.6 with interal 0.1 automatically with maximum validation Reward-Pass@2 scores. The optimal design choices outlined in §3, combined with our adaptive exploration strategy, form our final recipe for multimodal self-evolving training for reasoning, M-STAR.

**Results:** Table 4 presents the results of our final approach. By incorporating the dynamics of Reward-Pass@2, which balances both exploration and exploitation, our final recipe achieves the highest results, with 59.5% on the OOD test and 45.9% on the in-domain test. In contrast, models that only monitor Pass@K show diminished performance on both benchmarks. This reinforces the validity of our training design, demonstrating that an effective self-evolving training process requires a careful balance of both exploration and exploitation, as facilitated by the reward model. We also plot how the Pass@K and Reward-Pass@2 changes for M-STAR (*Reward-Pass@2*). To align with training, we show the metrics corresponding to the selected temperature in each iteration (see Appendix E for others). Figure 5 shows that compared with static strategy to choose a fixed temperature over the whole process, tuning it automatically mitigate the regression of Pass@K to help maintain the exploration ability. Besides, the Reward-Pass@2 is also generally higher than before. These further highlight the necessity to monitor the dynamics during training and adjust accordingly.

## 5 CONCLUSION

We dive into the self-evolving training for multimodal reasoning. Three static components are identified at first, namely the training method, reward model and the prompt variation. Through controlled experiments, we conclude a set of optimal design choices. On the other direction, we go deeper into the dynamics of self-evolving training to analyze the trade-off between exploitation and exploration. By monitoring the dynamics and adjusting key hyperparameters accordingly, we are able to further improve the model performance. We hope our work can provide insights and guidance for future research on self-evolving training for multimodal reasoning.

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

## A    COLLECTING WARMUP TRAINING DATA WITH CHAIN-OF-THOUGHT

Since our base model typically outputs the answer directly when responding to multimodal reasoning questions, during the warmup phase, we added additional instructions along with the input question, requiring the model to output the rationale. The instructions used in this process are as follows:

> **Extra instruction to guide CoT**
>
> Offer a comprehensive breakdown of your analytical process, detailing each step, the reasoning behind your decisions, and how you integrated various pieces of information, and put your answer at the end.

## B    HYPER PARAMETERS

We follow the training setup from Yao et al. (2024), using a learning rate of 1e-6 and a batch size of 128. A constant learning rate scheduler with a warmup ratio of 0.1 is applied. Input images are encoded using SigLIP SoViT-400m/14 (Zhai et al., 2023), and the visual tokens are compressed through a perceiver resampler structure with a single cross-attention layer. Additionally, each input image is sliced into a maximum of 9 segments, with each segment compressed into 96 queries.

## C    TRAINING PROCESS REWARD MODEL (PRM)

To train our PRM, we first train another checkpoint (denoted as $\hat{\pi}_\theta^0$) on our CoT-augmented training data for a much longer period to make sure it fully converges.

Based on this model, we leverage Monte Carlo Rollut method (Wang et al., 2024) to collect the training data for PRM. Specially, we randomly pick 50K questions from the full training set, and sample 16 responses for each of them with $\hat{\pi}_\theta^0$. We de-duplicate these responses, and only keep at most 4 responses for each question. After that we randomly sample 50K question-response pairs from all the pairs, where we control the ratio of correct and wrong responses as 1:1, and the ratio of multi-choice and free-form question as 1:1 as well, to keep a balanced distribution.

To construct the labels of each step, we use $\hat{\pi}_\theta^0$ as the completer to complete the solution from the end of each step in one response. For the $k^{\text{th}}$ step, the step label is annotated as $\frac{1}{N} \sum_{j=1}^{N} \mathbb{1}(C_j(s^{\leq k}) = a^*)$, where $N(= 16)$ is the number of completion, $C_j$ is the $j$-th completion.

Based on the stepwise annotations, we train our PRM from $\hat{\pi}_\theta^0$. We initialize the linear reward model head as the average of the embeddings, and train with MSE loss on all tokens, where the label of each token is identical to the step end token. In experiments we freeze the visual encoder as we find it brings a slight improvement.

## D    MEASURING RESPONSE RELATIVITY

To get a comprehensive understanding of how our PRM works as a re-ranker, we conduct a quantitative analysis using GPT4-o (`gpt-4o-2024-08-06`) to see how much a correct response is directly related to the query, e.g., does not contain irrelvant steps. The prompt we use is as follows:

> **Prompt for GPT4-o to annotate the relativity score**
>
> Given the image and a related question, you need to judge how a candidate solution is directly related to the question. You need to consider all its steps, and return a final value bewteen 1-10 as a overall score. Conclude your judgement at the end as "So the relativity score is X" where X is the score you give.
>
> [Question]
> {question}
>
> [Solution]
> {solution}

# E    MORE RESULTS FOR M-STAR

We plot the extra analysis results for M-STAR here. In Figure 6, we plot the changes of Pass@K and Reward-Pass@2 across different temperatures for M-STAR(*Reward-Pass@2*) as a compliment to the adapative adjustion mentioned in §4.2. We can see that acroos all selected temperatures, the exploration ability reflected by Pass@K does not regress continuously, and the Reward-Pass@2 reaches its peak more quickly, compared with training without the monitor of dynamics.

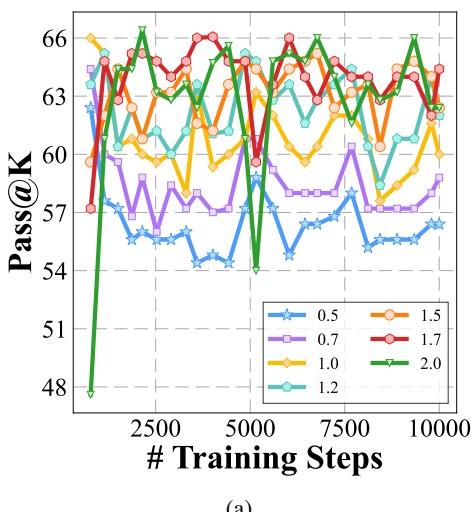
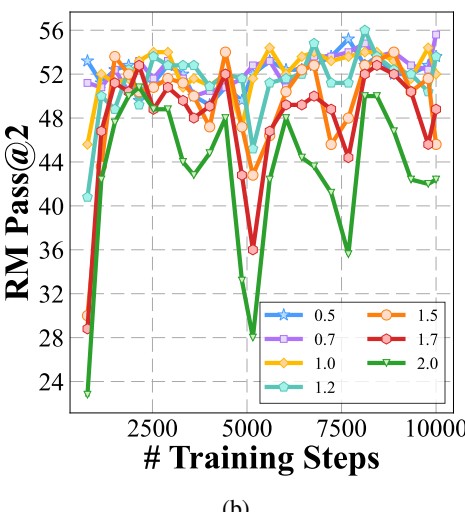

(a)                                                                          (b)

Figure 6: **(a)**:Pass@K changes during the training of M-STAR (*Reward-Pass@2*); **(b)**: :Reward-Pass@2 changes during the training of M-STAR (*Reward-Pass@2*). We pick 7 different temperatures.

# F    FULL RESULTS FOR MATHVISTA

To comprehensively evaluate the impact of different strategies for components in self-evolving training, we present the full results of MathVista, enabling a more detailed analysis. Instead of focusing solely on mathematical word problems (as one may be mislead by its name), MathVista actually encompasses a diverse set of reasoning-related tasks for LMMs, including figure question answering, visual question answering, science question answering, and more. As shown in Table 5, the overall performance corroborates our findings in § 3 and § 4, using three different models across three scales. The results demonstrate that the continuous self-evolving training method outperforms other self-evolving training approaches and simple SFT. Additionally, employing PRM as a Re-Ranker further enhances the performance of self-evolving training. Moreover, adjusting training dynamics provides additional performance gains, underscoring the importance of monitoring the training dynamics between exploration and exploitation during self-evolving training.

In addition to overall performance, we observe that self-evolving training based on larger models yields more comprehensive improvements across various sub-tasks. For instance, MiniCPMV-2.5 (8B), utilizing our optimal strategy and final recipe, achieves the best performance in 11 out of 12 sub-tasks, while Phi-3.5-Vision (4B) leads in 8 out of 12 sub-tasks. In contrast, the smaller model, InternVL2-2B, shows significant improvements primarily in math-related tasks. We speculate that this is because the training queries contain many math-related problems. Consequently, the smaller model struggles to generalize its learned abilities across different domains as effectively as the larger models, such as MiniCPMV-2.5 and Phi-3.5-vision.

# G    GENERALIZATION OF M-STAR

To further investigate how well M-STAR generalizes to benchmarks other than MathVista along, we select four extra multi-modal benchmarks focus on reasoning as well: M3CoT (Chen et al., 2024b), MMStar (Chen et al., 2024a), MMBench (Dev set, v1.1) (Liu et al., 2025), AI2D (Kembhavi et al.,

Table 5: Full analysis of MathVista. Task types: FQA: figure question answering, GPS: geometry problem solving, MWP: math word problem, TQA: textbook question answering, VQA: visual question answering. Mathematical reasoning types: ALG: algebraic reasoning, ARI: arithmetic reasoning, GEO: geometry reasoning, LOG: logical reasoning, NUM: numeric commonsense, SCI: scientific reasoning, STA: statistical reasoning.

| Model | ALL | FQA | GPS | MWP | TQA | VQA | ALG | ARI | GEO | LOG | NUM | SCI | STA |
|---|---|---|---|---|---|---|---|---|---|---|---|---|---|
| *MiniCPMV-2.5* | | | | | | | | | | | | | |
| MiniCPMV-2.5 | 52.4 | 59.2 | 44.7 | 50.5 | 53.8 | 48.0 | 42.7 | 46.5 | 46.0 | **29.7** | 36.1 | 56.7 | 60.1 |
| +warmup | 52.8 | 58.4 | 47.1 | 57.0 | 53.8 | 45.8 | 45.5 | 49.6 | 48.5 | 16.2 | 31.9 | 53.3 | 62.8 |
| SFT | 54.7 | 58.7 | 50.5 | 56.5 | 55.7 | 50.8 | 47.0 | 49.0 | 51.0 | 18.9 | 43.1 | 58.2 | 57.5 |
| Iterative RFT | 55.7 | 59.1 | 49.5 | 65.6 | 55.1 | 48.0 | 47.3 | 53.8 | 50.6 | 16.2 | 37.5 | 55.7 | 65.1 |
| Rest$^{EM}$ | 55.1 | 58.0 | 49.5 | 64.5 | 55.1 | 47.5 | 47.7 | 53.8 | 50.2 | 16.2 | 38.2 | 56.6 | 63.5 |
| Cont. optim. | 57.2 | 57.6 | 56.3 | 65.1 | 57.0 | 49.7 | 52.0 | 54.4 | 56.1 | 10.8 | 36.1 | 60.7 | 65.5 |
| +PRM Re-Rank | $59.2_{\uparrow 6.4}$ | $59.1_{\uparrow 0.7}$ | $\mathbf{61.1}_{\uparrow 14}$ | $\mathbf{68.3}_{\uparrow 11.3}$ | $55.1_{\uparrow 1.3}$ | $51.4_{\uparrow 5.6}$ | $54.8_{\uparrow 9.3}$ | $55.2_{\uparrow 5.6}$ | $\mathbf{60.3}_{\uparrow 11.8}$ | $10.8_{\downarrow 5.4}$ | $43.1_{\uparrow 11.2}$ | $59.0_{\uparrow 5.7}$ | $66.5_{\uparrow 3.7}$ |
| M-STAR | $\mathbf{59.5}_{\uparrow 6.7}$ | $\mathbf{59.5}_{\uparrow 1.1}$ | $59.1_{\uparrow 12}$ | $65.6_{\uparrow 8.6}$ | $\mathbf{58.9}_{\uparrow 5.1}$ | $\mathbf{54.2}_{\uparrow 8.4}$ | $\mathbf{54.5}_{\uparrow 9}$ | $\mathbf{56.7}_{\uparrow 7.1}$ | $58.2_{\uparrow 9.7}$ | $10.8_{\downarrow 5.4}$ | $43.1_{\uparrow 11.2}$ | $\mathbf{61.5}_{\uparrow 8.2}$ | $\mathbf{69.1}_{\uparrow 6.3}$ |
| *Phi-3.5-vision* | | | | | | | | | | | | | |
| Phi-3.5-vision | 46.5 | **58.7** | 36.5 | 36.0 | 56.3 | 41.9 | 39.5 | 38.8 | 36.4 | 16.2 | 34.0 | **60.7** | 62.8 |
| +warmup | 49.3 | 55.8 | 42.8 | 53.2 | 55.1 | 38.0 | 43.1 | 44.8 | 43.9 | 8.1 | 33.3 | 59.0 | 62.5 |
| SFT | 49.5 | 53.9 | 52.9 | 52.7 | 49.4 | 35.8 | 47.3 | 41.4 | 51.5 | **32.4** | 33.3 | 56.6 | 57.5 |
| Iterative RFT | 50.2 | 58.4 | 41.4 | 50.0 | 55.7 | 43.0 | 42.0 | 43.9 | 41.8 | 10.1 | 41.7 | 58.2 | 65.0 |
| Rest$^{EM}$ | 50.5 | 56.8 | 46.6 | 49.5 | **58.9** | 39.7 | 47.0 | 43.3 | 45.6 | 18.9 | 34.7 | 61.5 | 63.5 |
| Cont. optim. | 51.1 | 56.1 | 44.8 | 55.9 | 52.5 | 40.2 | 46.6 | 46.6 | 45.9 | 8.1 | 34.7 | 61.5 | 64.5 |
| +PRM Re-Rank | $53.2_{\uparrow 3.9}$ | $56.9_{\uparrow 1.1}$ | $51.9_{\uparrow 9.1}$ | $\mathbf{60.8}_{\uparrow 7.6}$ | $55.1_{0}$ | $39.7_{\uparrow 1.7}$ | $48.8_{\uparrow 5.7}$ | $46.2_{\uparrow 1.4}$ | $50.6_{\uparrow 6.7}$ | $5.4_{\downarrow 2.7}$ | $41.7_{\uparrow 8.4}$ | $59.8_{\uparrow 0.8}$ | $65.1_{\uparrow 2.6}$ |
| M-STAR | $\mathbf{54.5}_{\uparrow 5.2}$ | $56.9_{\uparrow 1.1}$ | $\mathbf{56.7}_{\uparrow 13.9}$ | $57.5_{\uparrow 4.3}$ | $55.1_{0}$ | $\mathbf{44.7}_{\uparrow 6.7}$ | $\mathbf{53.4}_{\uparrow 10.3}$ | $\mathbf{48.4}_{\uparrow 3.6}$ | $\mathbf{55.2}_{\uparrow 11.3}$ | $5.4_{\uparrow 2.7}$ | $\mathbf{42.4}_{\uparrow 9.1}$ | $56.6_{\downarrow 2.4}$ | $\mathbf{65.8}_{\uparrow 3.3}$ |
| *InternVL2-2B* | | | | | | | | | | | | | |
| InternVL2-2B | 46.4 | 53.2 | 45.2 | 33.3 | 50.0 | **48.0** | 41.6 | 41.4 | 43.1 | 10.8 | 25.7 | **55.7** | 59.8 |
| +warmup | 47.6 | 52.4 | 54.8 | 46.2 | 43.7 | 36.9 | 48.8 | 40.5 | 52.3 | **16.2** | 24.3 | 50.0 | 58.8 |
| SFT | 41.9 | 37.5 | 40.4 | 49.5 | 32.3 | 50.8 | 36.3 | 45.9 | 39.3 | 16.2 | **38.9** | 38.5 | 38.5 |
| Iterative RFT | 47.5 | 49.8 | **57.7** | 52.1 | 41.8 | 32.4 | 50.5 | 40.8 | 55.2 | 2.7 | 25.0 | 42.6 | 57.8 |
| Rest$^{EM}$ | 47.9 | 49.4 | 54.8 | 51.1 | **51.3** | 31.3 | 51.2 | 39.4 | 53.1 | 10.8 | 25.7 | 50.8 | 57.5 |
| Cont. optim. | 48.4 | **53.2** | 50.5 | 56.5 | 40.5 | 37.4 | 44.8 | 41.6 | 47.7 | 5.4 | 34.7 | 45.1 | **60.8** |
| +PRM Re-Rank | $48.8_{\uparrow 1.2}$ | $52.0_{\downarrow 0.4}$ | $55.8_{\uparrow 1.9}$ | $52.1_{\uparrow 5.9}$ | $45.6_{\uparrow 1.9}$ | $35.2_{\downarrow 1.7}$ | $50.2_{\uparrow 1.4}$ | $39.4_{\downarrow 1.1}$ | $55.2_{\uparrow 2.9}$ | $5.4_{\downarrow 10.8}$ | $33.3_{\uparrow 9}$ | $45.9_{\uparrow 4.1}$ | $60.5_{\uparrow 1.7}$ |
| M-STAR | $\mathbf{50.3}_{\uparrow 2.7}$ | $49.4_{\downarrow 3}$ | $57.2_{\uparrow 2.4}$ | $\mathbf{65.0}_{\uparrow 18.8}$ | $42.4_{\downarrow 1.3}$ | $35.2_{\downarrow 1.7}$ | $50.5_{\uparrow 1.7}$ | $\mathbf{47.0}_{\uparrow 6.5}$ | $\mathbf{56.1}_{\uparrow 3.8}$ | $13.5_{\downarrow 2.7}$ | $32.6_{\uparrow 8.3}$ | $45.9_{\downarrow 4.1}$ | $57.1_{\uparrow 1.7}$ |

Table 6: Performance of M-STAR compared with baselines and methods considering only static components. We highlight the relative improvement of M-STAR over the pre-evolved model, i.e., the "+warmup" row. For benchmark with suffix "-R", we follow Xu et al. (2024a) to remove some perception sub-tasks in them, to get the subsets that focus more on reasoning.

| | MathVista | M3CoT | MMStar-R | MMBench-R | AI2D | Average |
|---|---|---|---|---|---|---|
| MiniCPM-V-2.5 | 52.4 | 41.2 | 44.6 | 72.6 | 64.4 | 55.0 |
| + warmup | 52.6 | 47.8 | 45.1 | 76.9 | 65.9 | 57.7 |
| M-STAR | $\mathbf{59.5}_{\uparrow 6.9}$ | $\mathbf{48.7}_{\uparrow 0.9}$ | $\mathbf{50.7}_{\uparrow 5.6}$ | $\mathbf{79.9}_{\uparrow 3}$ | $\mathbf{69.1}_{\uparrow 3.2}$ | $\mathbf{61.6}_{\uparrow 3.9}$ |
| Phi-3.5-vision | 46.5 | 39.4 | 42.5 | 56.8 | 47.5 | 46.5 |
| + warmup | 49.3 | 46.5 | 44.2 | 70.9 | 65.5 | 55.3 |
| M-STAR | $\mathbf{54.5}_{\uparrow 5.2}$ | $\mathbf{51.3}_{\uparrow 4.8}$ | $\mathbf{48.8}_{\uparrow 4.6}$ | $\mathbf{73.6}_{\uparrow 2.7}$ | $\mathbf{67.9}_{\uparrow 2.4}$ | $\mathbf{59.2}_{\uparrow 3.9}$ |
| InternVL2-2B | 46.4 | 16.7 | 20.0 | 14.2 | 33.5 | 26.2 |
| + warmup | 47.6 | 45.6 | 41.8 | **68.8** | **60.0** | 52.8 |
| M-STAR | $\mathbf{50.3}_{\uparrow 2.7}$ | $\mathbf{47.1}_{\uparrow 1.5}$ | $\mathbf{42.0}_{\uparrow 0.2}$ | $67.3_{\downarrow 1.5}$ | $59.7_{\downarrow 0.3}$ | $\mathbf{53.3}_{\uparrow 0.5}$ |

2016). For MMStar and MMBench, we remove the perception sub-tasks in them to construct subsets focus more on reasoning. As shown in Table 6, models self-evolved with M-STAR consistently outperform both the base models and those trained with warmup across nearly all benchmarks. The only exception is InternVL2-2B, which underperforms on two benchmarks, aligning with the findings and speculations discussed in § F. Smaller models face greater challenges in generalizing beyond their training data, particularly on perception-intensive benchmarks like MMBench-R and AI2D. In contrast, larger models such as Phi-3.5-vision and MiniCPM-V-2.5 demonstrate significantly improved generalization, despite being trained with the same query set.

