# OpenReview forum: "Diving into Self-Evolve Training for Multimodal Reasoning"
_ICLR.cc/2025/Conference — Submitted to ICLR 2025_

### Official Review · Reviewer_wMNX · 2024-10-30

**Soundness:** 2
**Presentation:** 3
**Contribution:** 2
**Rating:** 5
**Confidence:** 3

**Summary:**

The paper analyses the key components of self-evolving training procedure for Multimodal Large Language Models (MLLMs) with the aim of getting new insights into the reasoning capabilities of MLLMs. In particular, three fundamental aspects of self-evolving learning, i.e., training method, reward model, and prompt variation are examined in a series of experiments involving MiniCPM-V 2.5 model. Furthermore, the authors delve into the dynamics of the self-evolution process by means of monitoring four metrics, representative for the analysed process.

**Strengths:**

1) The paper is clearly written, easy to follow, and its underlying theme is well motivated.
2) The experiments are well-designed and formally correct.
3) The considered topic is related to MLLMs, and as such is potentially of interest to the broad subset of the ICLR community.

**Weaknesses:**

1) The main problem is limited experimental evaluation with respect to the number of MLLMs employed in the study. The authors present the results for one particular MLLM with not discussion about how the outcomes generalize to other models. This generalization is obviously a critical issue.
2) Also, the authors consider only one dataset in their experiments. There are quite many datasets devoted to verifying abstract reasoning abilities of ML models, including MLLMs. Having a more diverse selection of these dataset (problem types) would be beneficial.

**Questions:**

1) Are the presented observations/conclusions related to MiniCPM-V 2.5 also valid for other MLLMs? If so, what is the foundation of such a claim.
2) A similar question regarding the validity of conclusions for other types of problems / other reasoning domains.
3) In Table 1 the results for In-Domain test samples are lower than those for OOD, which is surprising. What is the reason for a better OOD than ID performance?

---

> ### Author Response · Authors · 2024-11-23
> **Response to the Reviewer wMNX**
>
> Thank you for your time and effort in reviewing our paper. We appreciate your recognition of our writing, experimental design, and research topic. We will address your concerns below.
>
> > Weakness 1: The main problem is limited experimental evaluation with respect to the number of MLLMs employed in the study.
> > Question 1: Are the presented observations/conclusions related to MiniCPM-V 2.5 also valid for other MLLMs? If so, what is the foundation of such a claim.
>
> Thanks for your suggestion and question, we have already added two extra models, i.e., Phi-3.5-Vision-4B and InternVL2-2B in our experiments, please see our General Response for more details.
>
> > Weakness 2: Also, the authors consider only one dataset in their experiments. …. Having a more diverse selection of these dataset (problem types) would be beneficial.
> > Question 2: A similar question regarding the validity of conclusions for other types of problems / other reasoning domains.
>
> Thanks for your suggestion and question. We would like to clarify that the MathVista benchmark is already an aggregation of 31 datasets that cover visual question answering, logical reasoning, statistical reasoning and many other skills – it is not math-only but rather broad.Besides, we have already added four extra benchmarks, i.e. M3CoT, MMStar, MMBench and AI2D,  in our experiments, please see our General Response for more details.
>
> > Question 3: In Table 1 the results for In-Domain test samples are lower than those for OOD, which is surprising. What is the reason for a better OOD than ID performance?
>
> There are 3 main reasons:
>
> 1. Difficulty - our in-domain testset is sampled from the MathV360K dataset, which is constructed by difficulty-aware resampling to increase the ratio of hard questions and then augment the data further. So although it is an ID test set, its difficulty is inherently higher than Mathvista which lacks such  difficulty control.
> 2. Problem Type - Unlike MathVista, which predominantly requires simple numbers or multiple-choice options as the final answer, a significant portion of the ID MathV360K test set consists of open-ended questions that require full sentences as answers. This makes achieving an exact match— the default metric used—significantly more challenging for these types of questions.
> 3. Relative Improvement - a more reasonable view or our ID and OOD eval is to see the relative improvement. Compared with the base model and warmup-ed model, we have 25.2% improvement in ID testset, while only 0.2% on OOD testset.

---

> > ### Comment · Reviewer_wMNX · 2024-11-27
> >
> > I'd like to thank the authors for the rebuttal, which cleared some of my doubts.
> > I'm considering raising my score, though I still believe that the paper is below the acceptance threshold.

---

> > > ### Author Response · Authors · 2024-12-01
> > > **Response to the Reviewer wMNX**
> > >
> > > Thank you for your valuable feedback and for considering revising the score. In light of potential concerns regarding our motivation and contributions, we have provided further clarification in our general response. Please refer to the general response for more details.
> > >
> > > In summary, the key contributions of this paper include:
> > > - Conducting a pilot study to enhance multimodal reasoning, validated through comprehensive experiments,
> > > - Improving methodologies for self-evolving training,
> > > - Systematic investigations,
> > > - Developing resources (e.g., CoT data, PRM, and automatically annotated data).
> > >
> > > Our empirical experiments validate the effectiveness of these contributions step by step in detail.
> > >
> > > If you have any remaining concerns, please don't hesitate to let us know. Thank you!

---

> > > > ### Author Response · Authors · 2024-12-02
> > > > **Response to the Reviewer wMNX**
> > > >
> > > > Dear Reviewer wMNX,
> > > >
> > > > We hope this message finds you well. With only one day remaining for the review period, we are eager to know if our revisions and responses have adequately addressed your concerns. If you have any additional questions or require further clarification regarding our work, please do not hesitate to let us know. We would be more than happy to provide any additional information or address any remaining concerns.
> > > >
> > > > Thank you for your time and effort in reviewing our submission!

---

### Official Review · Reviewer_PDVC · 2024-11-02

**Soundness:** 3
**Presentation:** 3
**Contribution:** 3
**Rating:** 6
**Confidence:** 2

**Summary:**

This work focus on self-evolving training for multimodal reasoning. They mainly research on three components in self-evolving training, that is training method, reward model and prompt variation. By comparing performance of various configuration, which can be considered as a kind of grid-search, they finally determine a set of optimal design. Also, along the process, they gave some in-depth analysis and insight of different topics.

**Strengths:**

1. The layout of this paper is clear, the authors sperated the process into determination of three static components and stick to this layout by presenting each part in a reasonable order.
2. This work provide a comprehensive comparison of various configuration, which can serve as a reference for others working in this field. This work complement a study on self-evolving training method in multimodal reasoning area.
3. Along the process, the authors also provide some in-depth anasis in terms of diversed topic.

**Weaknesses:**

1. This work can be considered as a kind of grid-search process, and doesn't proposed new techniques in terms of methodology.

**Questions:**

1. Why is the order of focusing component be studied in the presented way? In the paper, authors first study in training method and determine a best configuration. Then they directly use this configuration for the study of reward model, finally the similar operation is performed to prompt variation. This way can be seen as a incomplete grid-search process, and the final optimal configuration could be different when they study in a different order. So it would be good for the authors to provide a reasonable explanation of this point.

---

> ### Author Response · Authors · 2024-11-23
> **Response to the Reviewer PDVC**
>
> Thank you for your time and effort in reviewing our paper. We appreciate your recognition of our paper's layout, comprehensive experimental comparisons, and in-depth analysis. We will address your concerns below.
>
> > Weakness 1: This work can be considered as a kind of grid-search process, and doesn't proposed new techniques in terms of methodology.
>
> We would like to highlight that our main contribution is to view self-improving algorithms from the lens of RL and identify several important aspects, then we study good practices of these empirically to present an effective training recipe. We believe that the empirical contributions are meaningful and helpful to help practitioners and significant enough.
>
> > Question 1: Why is the order of focusing component be studied in the presented way? … So it would be good for the authors to provide a reasonable explanation of this point.
>
> This is a valid point. For this type of work, it is typically challenging and computationally expensive to perform a full grid search across all possible combinations of factors in a training approach. As a result, the analysis is often conducted by examining one aspect at a time while keeping the others fixed at reasonable configurations to reduce computational cost. This approach is also commonly used in previous works [1,2]. While this method may not guarantee finding the optimal combination of factors, it is generally sufficient to identify a relatively effective training recipe that can benefit both researchers and practitioners.
> Regarding the specific order in which we studied the training method, reward model, and prompt variation, we chose this order based on the following reasoning: the training method is a fundamental and general aspect, and we expected that a successful training method should be broadly applicable across different settings. Therefore, we first focused on determining the training method. Next, our exploration of prompt variation involved unlabeled prompts that relied on reward signals provided by the reward models. Consequently, we studied reward models before investigating prompt variation.
>
> ---
>
> [1] Unpacking DPO and PPO: Disentangling Best Practices for Learning from Preference Feedback; NeurIPS 2024
>
> [2] What matters when building vision-language models?; NeurIPS 2024

---

> ### Author Response · Authors · 2024-12-01
> **Response to the Reviewer PDVC**
>
> Thank you once again for your valuable feedback and suggestions! We sincerely hope that our previous responses have addressed your concerns. If there are any remaining questions or issues, please don’t hesitate to let us know—we will do our utmost to address them further. If you feel that your concerns have been fully resolved, we would be truly grateful if you could consider revisiting your review score. We genuinely hope we’ve resolved everything to your satisfaction, but please don’t hesitate to reach out if there’s anything else we can assist you with!

---

> > ### Comment · Reviewer_PDVC · 2024-12-02
> >
> > I appreciate authors' effort in addressing my concerns. I will maintain my score.

---

> > > ### Author Response · Authors · 2024-12-02
> > >
> > > Thank you for your response! If you have any further questions or need additional information, please don’t hesitate to reach out to us.

---

### Official Review · Reviewer_rZP2 · 2024-11-03

**Soundness:** 3
**Presentation:** 3
**Contribution:** 2
**Rating:** 5
**Confidence:** 4

**Summary:**

This paper explores self-evolving training for enhancing multimodal reasoning in large multimodal models (LMMs), focusing on training without chain-of-thought annotations. The authors investigate three primary factors influencing the effectiveness of this training approach: training methods, reward model design, and prompt variation. They introduce a dynamic framework, M-STAR, built on MiniCPM-V 2.5, to optimize these factors. Key contributions include establishing best practices for each component, implementing a reward model to enhance response selection, and proposing an automatic temperature adjustment mechanism to balance exploration and exploitation during training.

**Strengths:**

1. Relevance: The paper addresses an important and timely problem—enhancing reasoning in large multimodal models without chain-of-thought annotations.

2. Quality: The paper demonstrates high technical quality in its systematic breakdown of training configurations, use of an adaptive exploration-exploitation strategy, and ablation studies.

3. Clarity: The paper is well-organized, with each section logically following from the last, making it easy to understand the flow from problem identification to solution.

**Weaknesses:**

1. Claim: The CoT warm-up phase conflicts with the paper’s definition of annotation/CoT-free self-evolving training. It is essential to clarify this reliance and provide a stronger rationale for the warm-up setting.

2. Empirical results: The paper primarily evaluates MathVista, a single multimodal reasoning benchmark focused on math-based tasks. To show broader applicability, the paper would benefit from additional benchmarks such as VQA or other scientific QA. Besides, there is only one LLM used in all experiments, so it's hard to justify whether the benefit of the method can be transferred to other architectures.

3. Theoretical analysis: Since self-evolving training shares similarities with RL, a theoretical analysis comparing these methods could clarify the unique aspects of M-STAR, such as optimality, stability, and guarantees. Also, the paper lacks a hypothesis-driven structure that ties the findings to the central research question, which appears more like a tech report rather than a research paper.

4. Contribution: While the paper has explored different configurations exhaustively, the overall contribution is vague given its lack of theoretical grounding and limitations in the empirical study.

**Questions:**

1. Ablation studies showing the model’s performance with and without the CoT phase to quantify its impact on results.

2. What's the computation cost of the proposed method compared to baselines?

3. Can the best configurations explored in this study be applied to other task domains without losing their effectiveness?

---

> ### Author Response · Authors · 2024-11-23
> **Response to Reviewer rZP2 (1/2)**
>
> Thanks for your time and effort in reviewing our paper. We appreciate your recognition of the relevance, the high technical quality and the writing clarity. We will address each of your concern as follows:
>
> > Weakness 1: Claim: The CoT warm-up phase conflicts with the paper’s definition of annotation/CoT-free self-evolving training. It is essential to clarify this reliance and provide a stronger rationale for the warm-up setting.
>
> We would like to clarify that our paper only claims annotation-free, but not CoT-free. Our paper studies multimodal reasoning where CoT is necessary. Since human-annotated CoT data is scarce for multimodal reasoning, our approach does not use any human-annotated CoT data – In the warmup stage, the training data is synthesized from prompting the base model (policy model) itself, and neither is extra annotation needed nor other models are involved, so we do not think this warmup stage violates any self-evolving claims in this paper
> .
> > Weakness 2: Empirical results: The paper primarily evaluates MathVista, a single multimodal reasoning benchmark focused on math-based tasks. To show broader applicability, the paper would benefit from additional benchmarks such as VQA or other scientific QA. Besides, there is only one LLM used in all experiments, so it's hard to justify whether the benefit of the method can be transferred to other architectures.
>
> 1. Regarding the evaluation datasets, we would like to clarify that the MathVista benchmark is already an aggregation of 31 datasets that cover geometry reasoning, logical reasoning, statistical reasoning and many other skills – it is not math-only but rather broad. It is just that we only reported the average score on MathVista in the submission. To give more evidence on our performance on multiple tasks, we report more results on each subtask of MathVista, as shown in the General Response as well as in Appendix F of the updated PDF. We can see that most of the subtasks show improvements as we improve the configuration step by step. The final recipe, MStar, also leads to optimal results for most of them.
>
> 2. Besides MathVista, to better address your concern, we also add several new multimodal reasoning benchmarks, including M3CoT[1], MMStar[2], MMBench[3] and AI2D[4],  as shown in the General Response as well as in Appendix G. We can see for all these benchmarks, M-Star gives consistent improvements.
>
> 3. For your comment on using only one model, we have also added new results on two new models, Phi-3.5-vision-4B and InternVL2-2B. The results are included as well in the General Response and Appendix F, G of the updated PDF. Similar to what we have observed on MiniCPM-V-2.5-8B, our findings hold for them in general, and M-Star can boost their performance on a wide range of benchmarks.
>
> > Weakness 3: Theoretical analysis: Since self-evolving training shares similarities with RL, a theoretical analysis comparing these methods could clarify the unique aspects of M-STAR, such as optimality, stability, and guarantees.
>
> Thank you for your feedback and for raising these important points. We would like to further clarify our motivation and claims.
>
> In this paper, we start from the observation that self-evolving training can be subsumed as a general reinforcement learning (RL) training process, as described in Eq. 1. However, prior works on [1, 2, 3], have primarily treated self-evolving training as an iterative training paradigm. These works have largely overlooked the deeper connection between self-evolving training and the reinforcement learning framework. Based on this observation, we aim to investigate whether the three key factors within reinforcement learning—online/offline training, reward model, and prompt variation—can contribute to improving self-evolving training. Additionally, we analyze the training dynamics of the self-evolving training process, focusing on the balance between exploration and exploitation. We believe that these perspectives set our paper unique compared to previous self-evolving training works.
>
> ---
>
> [1] STaR: Bootstrapping Reasoning With Reasoning; NeurIPS 2022
>
> [2] Beyond Human Data: Scaling Self-Training for Problem-Solving with Language Models; TMLR 2024
>
> [3] V-STaR: Training Verifiers for Self-Taught Reasoners; COLM 2024

---

> ### Author Response · Authors · 2024-11-23
> **Response to Reviewer rZP2 (2/2)**
>
> > Weakness 3: Also, the paper lacks a hypothesis-driven structure that ties the findings to the central research question, which appears more like a tech report rather than a research paper.
>
> Our central research question is to investigate several key components of the self-improving algorithm, as motivated and outlined in the earlier sections of the paper (Section 2, Lines 120–131). Different papers may adopt different writing structures, and in this work, we follow a structure that systematically explores various aspects and draws conclusions step by step through our experiments. We do not think that this writing style constitutes a significant flaw in the paper. For instance, many prior works have also adopted similar structures to empirically study best practices, such as [1,2].
>
> ---
>
> [1] Unpacking DPO and PPO: Disentangling Best Practices for Learning from Preference Feedback; NeurIPS 2024
>
> [2] What matters when building vision-language models?; NeurIPS 2024
>
>
> > Weakness 4: Contribution: While the paper has explored different configurations exhaustively, the overall contribution is vague given its lack of theoretical grounding and limitations in the empirical study.
>
> Our main contribution is to approach self-improving algorithms through the lens of reinforcement learning (RL), identify several critical aspects, and empirically study best practices to propose an effective training recipe. While we did not provide deep theoretical grounding for many of the phenomena observed, we argue that our empirical findings are both meaningful and valuable for practitioners, making the contribution significant. In particular, the additional models and evaluations presented in the General Response further strengthen this contribution.
>
> > Question 1: Ablation studies showing the model’s performance with and without the CoT phase to quantify its impact on results.
>
> We would like to emphasize that the warmup phase with CoT training is essential because we observed that most open-weight multimodal models, such as Llava and MiniCPM-V, rarely generate CoT reasoning and typically only produce a short final answer. For self-evolving training in reasoning tasks, the key improvements are achieved by enhancing the reasoning process, specifically the CoT reasoning – it is just how this setting is defined. This follows the standard setting adopted by nearly all prior works in this area [1,2,3,4]. Thus, conducting self-evolving reasoning training when the model cannot generate CoT reasoning would be largely ineffective. This is precisely why we include the CoT warmup phase in the first place. As stated in our response to Weakness 1 of the reviewer, we do not rely on any additional annotations or external resources during the warmup phase. Therefore, we believe there is no issue or limitation associated with this approach.
>
> ---
>
> [1] Scaling relationship on learning mathematical reasoning with large language models; arXiv 2308.01825
>
> [2] STaR: Bootstrapping Reasoning With Reasoning; NeurIPS 2022
>
> [3] V-STaR: Training Verifiers for Self-Taught Reasoners; COLM 2024
>
> [4] Beyond Human Data: Scaling Self-Training for Problem-Solving with Language Models; TMLR 2024
>
> > Question 2: What's the computation cost of the proposed method compared to baselines?
>
> Admittedly, self-evolving training requires more computational resources compared to the SFT baseline. This is a common characteristic of self-evolving training approaches [1, 2] and reinforcement learning with human feedback (RLHF) [3, 4]. The additional computation cost mainly comes from extra data generation processes of these RL-like approaches and typically more training steps than SFT. However, when comparing different variations of self-evolving training, the computational costs are largely similar, as we are all training for the same number of steps with the same batch size for all self-evolving runs.
>
> ---
>
> [1] STaR: Bootstrapping Reasoning With Reasoning; NeurIPS 2022
>
>
> [2] ReST-MCTS*: LLM Self-Training via Process Reward Guided Tree Search; NeurIPS 2024
>
>
> [3] TÜLU 3: Pushing Frontiers in Open Language Model Post-Training; Tech Report
>
> [4] The Llama 3 Herd of Models; arxiv 2407.21783
>
> > Question3: Can the best configurations explored in this study be applied to other task domains without losing their effectiveness?
>
> As shown in the general response, we have applied our training recipe to other two models, Phi-3.5-Vision-4B and InternVL2-2B as well and evaluated these models on several added benchmarks and tasks. The results demonstrate that our best configurations are effective in most of these settings.

---

> > ### Comment · Reviewer_rZP2 · 2024-11-27
> >
> > I appreciate the detailed replies and new results from the authors. Most of my specific questions have been addressed. However, my concerns remain regarding the novelty and contribution of the paper. More specifically, the paper focuses more on the empirical side, without providing convincing theoretical backing or demonstrating strong results compared to existing work. While the findings could be very useful for some practical use cases, I would recommend that the paper (in its current state) either dive deeper into the problem of self-supervised reasoning or target a more practically focused venue or workshop.

---

### Official Review · Reviewer_tYHA · 2024-11-04

**Soundness:** 3
**Presentation:** 3
**Contribution:** 2
**Rating:** 5
**Confidence:** 4

**Summary:**

The paper, "Diving into Self-Evolving Training for Multimodal Reasoning," explores the enhancement of reasoning abilities in Large Multimodal Models (LMMs) through self-evolving training, a method where models iteratively improve by learning from their own outputs. The absence of multimodal chain-of-thought annotated data has led to this innovative approach. The study identifies and systematically examines three critical factors—Training Method, Reward Model, and Prompt Variation—that influence the effectiveness of training. The authors present a comprehensive analysis and establish best practices for each factor within a newly proposed framework named M-STAR (Multimodal Self-evolving Training for Reasoning), built on MiniCPM-V 2.5. This framework achieved a significant improvement in accuracy on the MathVista dataset, demonstrating its efficacy. The paper also explores the dynamics of self-evolution and introduces an automatic mechanism to balance model exploration and exploitation, further enhancing performance.

**Strengths:**

(1) The paper introduces an original framework, M-STAR, for self-evolving training in multimodal reasoning. This approach is particularly innovative as it leverages the model's own outputs for iterative improvement, a method relatively underexplored in the context of multimodal reasoning. Additionally, the focus on three specific components (Training Method, Reward Model, and Prompt Variation) for optimizing training presents a novel angle for investigation.

(2) The paper is well-structured and clearly written. The authors effectively communicate complex ideas, such as the dynamics of self-evolution and the implementation of an automatic balancing mechanism during training. The systematic breakdown of each key factor and the subsequent analysis make the paper accessible to readers with varying levels of expertise in the field.

**Weaknesses:**

(1) The motivation or evidence behind the importance of the three components: the training method, the use of the reward model, and the prompt variation is insufficiently substantiated. The authors need to provide more detailed justification or empirical evidence to support the significance of these components in the context of multimodal reasoning.

(2) The ablation experiments are solely based on a single model: MiniCPM-V-2.5, and two datasets from the Math domain. It would be beneficial to explore the effects of different model sizes (understanding the constraints of increased training time with larger models, experimenting with smaller models could be insightful) and datasets from varied domains such as code generation to generalize the findings.

(3) The settings of the ablation studies focus primarily on minor hyperparameter adjustments, leading to conclusions that align with conventional expectations. It is recommended that the authors delve deeper into algorithmic comparisons. For instance, contrasting with techniques like Supervised Fine-Tuning (SFT), Reinforcement Learning from Human Feedback (RLHF), or Differentiable Prompt Optimization (DPO), as well as exploring different training methodologies (e.g., multi-training stages) or network architecture designs (e.g., with different multimodal encoders) could provide more robust insights.

**Questions:**

(1) Could the authors elaborate on the specific motivations or additional evidence that underscore the criticality of the training method, reward model, and prompt variation in enhancing multimodal reasoning? A deeper understanding or empirical backing could significantly strengthen the paper's foundation.

(2) Have the authors considered expanding the ablation studies to include a broader range of model sizes, including smaller ones, despite the acknowledged increased training time with larger models? Additionally, could the use of datasets from different domains, such as code generation, provide more comprehensive insights into the model's capabilities and limitations?

(3) In terms of algorithmic comparisons and training methodologies, could the authors provide a comparative analysis with other prevalent techniques like SFT, RLHF, DPO, or different training stages and network structures? Such comparisons could offer a clearer differentiation and possibly highlight the advantages or limitations of the proposed M-STAR framework in a broader context.

(4) In the section "MONITORING THE TRAINING DYNAMICS," it is observed that nearly all training reaches its peak performance quickly (within < 2500 steps), after which the model's performance tends to decline as training progresses. Does this suggest that the base self-evolving training configuration might not be optimally set? For instance, issues such as an overly small dataset, an excessively large model, or inappropriate regularization settings could be contributing factors. How do these factors influence the conclusions drawn from the training baseline, and could this impact the accuracy of the study's outcomes?

(5) Beyond assessing the correctness of results in the Math domain, should the evaluation of the model's outputs also consider other dimensions? For instance, evaluating aspects such as the interpretability, robustness, or even the creativity of the responses could provide a more holistic view of the model's capabilities in multimodal reasoning. How do the authors envision incorporating these additional evaluation metrics into their framework?

---

> ### Author Response · Authors · 2024-11-23
> **Response to the Reviewer tYHA (1/3)**
>
> Thanks for your time and effort in reviewing our paper. We appreciate your recognition of our investigation of self-evolution in multimodal reasoning, and our systematic breakdown of each key factor in it. We will address each of your concern as follows:
>
> > Weakness1: The motivation or evidence behind the importance of the three components: the training method, the use of the reward model, and the prompt variation is insufficiently substantiated.
> > Question 1: Could the authors elaborate on the specific motivations or additional evidence that underscore the criticality of the training method, reward model, and prompt variation in enhancing multimodal reasoning?
>
> Thank you for the suggestion. Our selection of these three components as the main research focus in this paper is primarily based on our modeling of self-evolving training within a general RL framework (as discussed in Section 2.), which has been similarly modeled in many RL-related papers [1, 2, 3, 4, 5].
>
>
> These three selected components, training methods, reward models, and prompt variation, have been recognized as essential and studied in many similar works, particularly in text-only settings. Regarding the training method, [3, 4, 5] discovered that factors such as online/offline updates and on-policy/off-policy rollouts in the training algorithm can greatly impact model performance. For the reward model, [6,7,8] investigated the impact of different reward function, reward model on the training results; and for prompt variation, [1] explored how different prompts in RL would affect the alignment RLHF training results. However, most of previous works except [1] only considers one factor instead of comprehensively investigating all these factors. Meanwhile, none of previous works study how these important factors in RL training would affect the training results of self-evolving .Since self-evolving training follows the general RL framework, we identify the three important and popular factors inside RL training following previous works in this direction. Then we systematically study these factors in a more complex and rarely discussed domain, multimodal reasoning.
>
>
> We also acknowledge that other factors, such as model architecture, hyper-parameter configurations (e.g., learning rate, batch size), and the specific updating algorithms, may influence the overall pipeline. However, some of these factors are orthogonal to our RL framework. Thus, we have made practical selections, such as using an advanced LMM , MiniCPM-V-2.5, for all ablations and keeping the learning rate and batch size unchanged.
>
> ---
>
> [1] Unpacking DPO and PPO: Disentangling Best Practices for Learning from Preference Feedback; NeurIPS 2024
>
> [2] Decoupling Exploration and Exploitation for Meta-Reinforcement Learning without Sacrifices; PMLR 2021
>
> [3] RLHF Workflow: From Reward Modeling to Online RLHF; TMLR, 2024
>
> [4] Direct Language Model Alignment from Online AI Feedback; ICML 2024
>
> [5] DeepSeekMath: Pushing the Limits of Mathematical Reasoning in Open Language Models; arxiv.2402.03300
>
> [6] V-STaR: Training Verifiers for Self-Taught Reasoners; COLM 2024
>
> [7] DeepSeek-Coder-V2: Breaking the Barrier of Closed-Source Models in Code Intelligence; arxiv 2406.11931
>
> [8] Rewarding Progress: Scaling Automated Process Verifiers for LLM Reasoning; arxiv 2410.08146

---

> > ### Author Response · Authors · 2024-11-23
> > **Response to the Reviewer tYHA (2/3)**
> >
> > > Weakness 2: The ablation experiments are solely based on a single model: MiniCPM-V-2.5, and two datasets from the Math domain. It would be beneficial to explore the effects of different model sizes (understanding the constraints of increased training time with larger models, experimenting with smaller models could be insightful) and datasets from varied domains such as code generation to generalize the findings.
> > > Question 2: It would be beneficial to explore the effects of different model sizes and datasets from varied domains
> >
> > Thanks for the suggestion and the question.
> >
> > 1. Regarding the evaluation datasets, we would like to clarify that the MathVista benchmark is already an aggregation of 31 datasets that cover geometry reasoning, logical reasoning, statistical reasoning and many other skills – it is not math-only but rather broad. It is just that we only reported the average score on MathVista in the submission. To give more evidence on our performance on multiple tasks, we report more results on each subtask of MathVista, as shown in the General Response as well as in Appendix F of the updated PDF. We can see that most of the subtasks show improvements as we improve the configuration step by step. The final recipe, MStar, also leads to optimal results for most of them.
> >
> > 2. Besides MathVista, to better address your concern, we also add several new multimodal reasoning benchmarks, including M3CoT[1], MMStar[2], MMBench[3] and AI2D[4],  as shown in the General Response as well as in Appendix G. We can see for almost all these benchmarks, M-Star gives consistent improvements.
> >
> > 3. We also add new results on two new models, Phi-3.5-vision-4B and InternVL2-2B. The results are included as well in the General Response and Appendix F, G of the updated PDF. Similar to what we have observed on MiniCPM-V-2.5-8B, our findings hold for them in general, and M-Star can boost their performance on a wide range of benchmarks.
> >
> >
> > > Weakness 3: It is recommended that the authors delve deeper into algorithmic comparisons. For instance, contrasting with techniques like Supervised Fine-Tuning (SFT), Reinforcement Learning from Human Feedback (RLHF), or Differentiable Prompt Optimization (DPO)....
> > > Question 3: could the authors provide a comparative analysis with other prevalent techniques like SFT, RLHF, DPO...
> >
> > Thank you for the suggestion. Comparing different loss functions and algorithms is indeed a non-trivial task. In this paper, we primarily focus on the key components of the iterative rejection fine-tuning approach, which is one of the most widely used and robust self-improving algorithms, as demonstrated in STaR [1] and Llama 3 [2]. Given the widespread adoption of iterative rejection fine-tuning, we argue that a thorough investigation of its critical components is significant enough. We leave a comprehensive comparison of different self-improving algorithms to future work.
> >
> > ---
> > [1] STaR: Bootstrapping Reasoning With Reasoning; NeurIPS 2022
> >
> > [2] The Llama 3 Herd of Models; arxiv 2407.21783
> >
> >
> > > Question 4: it is observed that nearly all training reaches its peak performance quickly (within < 2500 steps), after which the model's performance tends to decline as training progresses. Does this suggest that the base self-evolving training configuration might not be optimally set?
> >
> > We would like to clarify several points regarding the mentioned phenomenon:
> >
> > 1. The model’s performance reflected by greedy decoding does not decline, it increases first and saturates (Figure 3)
> >
> > 2. The declining trend in Figure 4(a)(b) is in terms of pass@K accuracy, this is due to the decline of model’s exploration ability when the model is trained on its own outputs, which is a known issue of self-improving as also observed in [1]
> >
> > 3. Quick saturation in self-improving training is very common [1,2,3], and it can stem from various factors, such as the limitations of the training set (our training set contains 180K examples, which is not overly small) and the quality of the reward models. Based on our analysis in Section 3.3, which highlights the challenges of obtaining an effective PRM verifier, we hypothesize that the reward model may play a significant role in causing this quick saturation. Actually, overcoming quick saturation and achieving scalable improvements is one of the most important challenges currently for self-improving algorithms and involves multiple research problems, which we leave as future work to study.
> >
> > ---
> >
> > [1] Progress or Regress? Self-Improvement Reversal in Post-training; arXiv 2407.05013
> >
> > [2] Entropic Distribution Matching in Supervised Fine-tuning of LLMs: Less Overfitting and Better Diversity; arXiv:2408.16673
> >
> > [3] Mitigating Tail Narrowing in LLM Self-Improvement via Socratic-Guided Sampling; arXiv:2411.00750

---

> > > ### Author Response · Authors · 2024-11-23
> > > **Response to the Reviewer tYHA (3/3)**
> > >
> > > > Question 5: Beyond assessing the correctness of results in the Math domain, should the evaluation of the model's outputs also consider other dimensions? For instance, evaluating aspects such as the interpretability, robustness, or even the creativity of the responses could provide a more holistic view of the model's capabilities in multimodal reasoning. How do the authors envision incorporating these additional evaluation metrics into their framework?
> > >
> > > Thanks for this suggestion. As we explained in the response to Weakness 2 & Question 2, Mathvista covers rather broad domains and we also added several other multimodal reasoning benchmarks to support our empirical effectiveness.
> > >
> > > Besides, we definitely agree with your suggestion that incorporating extra evaluation dimensions like interpretability, robustness or even creativity other than correctness only will make the analysis more comprehensive. However, there are no well-established benchmarks on multimodal reasoning to evaluate these aspects, and we have to admit that the reliable and automatic assessment of these metrics are challenging. For now, we only focus on question answering settings which are inherently designed to measure correctness and more suitable data are needed to evaluate these metrics which can be quite scarce in the multimodal domain.
> > >
> > > Nevertheless, we are still open to making the evaluation as comprehensive as possible, for instance involving other LMMs (like GPT4o) or fine-tuning some judge LMMs [1]  to automatically assess multiple dimensions, or even build a complex evaluation system [2, 3] for it. We believe although this may not be in the short-term scope of this paper, we will still pursue this in the long run.
> > >
> > > ---
> > >
> > > [1] Prometheus-Vision: Vision-Language Model as a Judge for Fine-Grained Evaluation; ACL 2024
> > >
> > > [2] Is Your Model Really A Good Math Reasoner? Evaluating Mathematical Reasoning with Checklist; arXiv 2407.08733
> > >
> > > [3] LiveBench: A Challenging, Contamination-Free LLM Benchmark; arXiv 2406.19314

---

> > > > ### Comment · Reviewer_tYHA · 2024-11-26
> > > >
> > > > Thank you for the detailed response, which addressed most of my concerns. After reviewing the other reviewers' comments and the rebuttal, I have decided to maintain my original score.

---

### Author Response · Authors · 2024-11-23
**General Response to the Reviewers (1/2)**

We thank all the reviewers for the comments! We have revised the PDF to reflect the reviewers' comments, and responded to each reviewer separately in the respective thread. Here we summarize the main revisions of the manuscript.

Adding new models (Appendix F): We introduce two additional models of different sizes: **Phi-3.5-vision** (4B)[1] and **InternVL2-2B** (2B)[2]. To provide a more comprehensive analysis, we leverage the full results from MathVista to study various training recipes in greater depth. It is important to note that MathVista is not solely a benchmark for math word problems; it is a comprehensive multimodal benchmark encompassing a wide range of reasoning tasks, including **visual question answering**, **figure-based question answering**, **science question answering**, **and more**. The results are briefly shown below and the detailed results can be found in Appendix F. (addressing Reviewer tYHA, rZP2, wMNX)

Adding new benchmarks (Appendix G): To further validate the effectiveness of our **M-STAR** recipe, we evaluate it on four additional benchmarks: **M3CoT**[3], **MMStar-R**[4], **MMBench-R**[5], and **AI2D**[6] (-R indicates the the selected reasoning subset. The results are briefly shown below and detailed in Appendix G as well, demonstrating that M-STAR significantly enhances the multimodal reasoning abilities of models across different sizes and benchmarks. (addressing Reviewer  tYHA, rZP2, wMNX)

More comprehensive report on a breakdown of different datasets in MathVista (here we report MiniCPMV-2.5, full results of other models are in Table 5, Appendix F):

| **Model**            | **ALL**        | **FQA**        | **GPS**          | **MWP**          | **TQA**        | **VQA**        | **ALG**        | **ARI**        | **GEO**          | **LOG**        | **NUM**        | **SCI**        | **STA**        |
|----------------------|----------------|----------------|------------------|------------------|----------------|----------------|----------------|----------------|------------------|----------------|----------------|----------------|----------------|
| MiniCPMV-2.5         | 52.4           | 59.2           | 44.7             | 50.5             | 53.8           | 48.0           | 42.7           | 46.5           | 46.0             | **29.7**       | 36.1           | 56.7           | 60.1           |
|     +warmup          | 52.8           | 58.4           | 47.1             | 57.0             | 53.8           | 45.8           | 45.5           | 49.6           | 48.5             | 16.2           | 31.9           | 53.3           | 62.8           |
| SFT                  | 54.7           | 58.7           | 50.5             | 56.5             | 55.7           | 50.8           | 47.0           | 49.0           | 51.0             | 18.9           | 43.1           | 58.2           | 57.5           |
| Iterative RFT        | 55.7           | 59.1           | 49.5             | 65.6             | 55.1           | 48.0           | 47.3           | 53.8           | 50.6             | 16.2           | 37.5           | 55.7           | 65.1           |
| RestEM    | 55.1           | 58.0           | 49.5             | 64.5             | 55.1           | 47.5           | 47.7           | 53.8           | 50.2             | 16.2           | 38.2           | 56.6           | 63.5           |
| Cont. optim.         | 57.2           | 57.6           | 56.3             | 65.1             | 57.0           | 49.7           | 52.0           | 54.4           | 56.1             | 10.8           | 36.1           | 60.7           | 65.5           |
|     + **PRM Re-Rank**     | 59.2 (**+6.4**)| 59.1 (**+0.7**)| **61.1 (+14.0)** | **68.3 (+11.3)** | 55.1 (**+1.3**)| 51.4 (**+5.6**)| 54.8 (**+9.3**)| 55.2 (**+5.6**)| **60.3 (+11.8)** | 10.8 (**-5.4**)| 43.1 (**+11.2**)| 59.0 (**+5.7**)| 66.5 (**+3.7**)|
| **M-STAR**           | **59.5 (+6.7)**| **59.5 (+1.1)**| 59.1 (**+12.0**)| 65.6 (**+8.6**)| **58.9 (+5.1)**| **54.2 (+8.4)**| **54.5 (+9.0)**| **56.7 (+7.1)**| 58.2 (**+9.7**)| 10.8 (**-5.4**)| **43.1 (+11.2)**| **61.5 (+8.2)**| **69.1 (+6.3)**|

---

> ### Author Response · Authors · 2024-11-23
> **General Response to the Reviewers (2/2)**
>
> Results with additional models and benchmarks:
>
> ### MiniCPM-V-2.5
> | Model               | MathVista      | M3CoT          | MMStar-R       | MMBench-R      | AI2D           | Average        |
> |---------------------|----------------|----------------|----------------|----------------|----------------|----------------|
> | MiniCPM-V-2.5       | 52.4           | 41.2           | 44.6           | 72.6           | 64.4           | 55.0           |
> |     + warmup        | 52.6           | 47.8           | 45.1           | 76.9           | 65.9           | 57.7           |
> | **M-STAR**           | **59.5** (**+6.9**)| **48.7** (**+0.9**)| **50.7** (**+5.6**)| **79.9** (**+3.0**)| **69.1** (**+3.2**)| **61.6** (**+3.9**)|
> ---
> ### Phi-3.5-vision
> | Model               | MathVista      | M3CoT          | MMStar-R       | MMBench-R      | AI2D           | Average        |
> |---------------------|----------------|----------------|----------------|----------------|----------------|----------------|
> | Phi-3.5-vision      | 46.5           | **39.4**       | 42.5           | 56.8           | 47.5           | 46.5           |
> |     + warmup        | 49.3           | 46.5           | 44.2           | 70.9           | 65.5           | 55.3           |
> | **M-STAR**           | **54.5** (**+5.2**)|  **51.3** (**+4.8**)| **48.8** (**+4.6**)| **73.6** (**+2.7**)| **67.9** (**+2.4**)| **59.2** (**+3.9**)|
> ---
> ### InternVL2-2B
> | Model               | MathVista      | M3CoT          | MMStar-R       | MMBench-R      | AI2D           | Average        |
> |---------------------|----------------|----------------|----------------|----------------|----------------|----------------|
> | InternVL2-2B        | 46.4           | 16.7           | 20.0           | 14.2           | 33.5           | 26.2           |
> |     + warmup        | 47.6           | 45.6           | 41.8           | **68.8**       | **60.0**       | 52.8           |
> | **M-STAR**           | **50.3** (**+2.7**)| **47.1** (**+1.5**)| **42.0** (**+0.2**)| 67.3 (-1.5)    | 59.7 (-0.3)    | **53.3** (**+0.5**)|
> ---
> [1] Phi-3 Technical Report: A Highly Capable Language Model Locally on Your Phone; arXiv 2404.14219
>
> [2] InternVL: Scaling up Vision Foundation Models and Aligning for Generic Visual-Linguistic Tasks; CVPR 2024
>
> [3] M3CoT: A Novel Benchmark for Multi-Domain Multi-step Multi-modal Chain-of-Thought; ACL 2024
>
> [4] Are We on the Right Way for Evaluating Large Vision-Language Models?; NeurIPS 2024
>
> [5] MMBench: Is Your Multi-modal Model an All-around Player?; ECCV 2024
>
> [6] A Diagram Is Worth A Dozen Images; ECCV 2016

---

### Author Response · Authors · 2024-11-25
**A Kind Reminder for Reading the Response**

Dear Reviewers,

We have carefully revised the paper and included additional results to address the valuable feedback you provided. As the rebuttal period is coming to a close, we kindly request if you could review our responses to ensure they adequately address your concerns. We would greatly appreciate that!

Thank you,
The authors

---

### Author Response · Authors · 2024-12-01
**Further Clarification on Our Motivation and Contributions**

## **General Response for Motivation and Contributions**

We thank all the reviewers once again for taking the time to review our work and for providing valuable feedback. We appreciate your insights and suggestions. However, we believe there are still some areas of misalignment regarding the motivation and contributions of our work. To address this, we would like to provide further clarification and highlight the key points.
The motivation behind our work lies at the intersection of two key areas: **multimodal reasoning** and **self-evolving training**. However, we noted that our contributions to multimodal reasoning is not  fully recognized.
Below, we detail our contributions within each domain:

### **Multimodal Reasoning**

1. This is the first work to equip Large Multimodal Models (LMMs) with CoT reasoning for a broad set of multimodal reasoning tasks via self-generation. Given the scarcity of high-quality CoT data in MM reasoning, many popular open-source LMMs, such as LLaVa, MiniCPM, and InternVL, do not demonstrate CoT reasoning abilities for reasoning tasks – they often directly produce the answer even though we used CoT prompts in our preliminary experiments.
2. We propose a novel and effective self-evolving training framework for multimodal reasoning, relying solely on the base model itself. This is the first work to study a complete self-evolving framework for multimodal reasoning blended with online learning, unlabeled prompts, and process reward models.
3. We train the first multimodal process reward model (PRM), which has not been explored in previous literature, and integrate it into self-evolving training. We will open-source all data collection and training pipeline

Finally, we emphasize that the effort behind this work is nontrivial. Unlike text-only LLMs, MM reasoning lacks adequate support and resources. We invested significant computation, effort, and iterative trials to develop a robust and effective pipeline in this domain, while also analyzing the impact of various contributing factors.

### **Self-Evolving Training**
1. Self-evolving training is effective but far from full explorations. So in this work, we
summarize key components for self-evolving training through the lens of RL and study them in a detailed manner and lead to a successful recipe.
2. We study the self-evolution dynamics from the “exploration & exploitation” trade-off and propose a novel dynamic monitor to enable adaptive exploration during the training process.

------

Overall, in this paper, we have contributed:
- A pilot study to enhance multimodal reasoning validated by comprehensive studies,
- Improved methodologies for self-evolving training,
- Systematic investigations,
- Constructed resources (e.g., CoT data, PRM and automatically annotated data).

Our empirical experiments substantiate the effectiveness of these contributions stey by step in detail.

---

### Meta-Review · Area_Chair_vCsE · 2024-12-21

**Metareview:**

The paper proposes M-STAR, used for training multi-modal reasoning into language models.
  * Assuming that the reader is familiar with basic RL-HF methods, the main new contributions consist of applying well-known techniques (e.g. Process-based Reward models, variants of self-training over correct synthetic data) over multi-modal benchmarks
  * Ablations were performed over different hyperparameter settings (e.g. sampling methods, temperature).
  * Results are shown for MathVista, although post-rebuttal, results over more multi-modal benchmarks and base models were provided.

The core issue of this paper is that we don't find any new or surprising conclusions, other than validating that previous techniques also work for multi-modal reasoning. This issue is nearly unanimously agreed by the reviewers, and it's not clear how the setting being multimodal makes the RL training any different from regular text-only training, i.e. all the results are what we'd expect from general RL for LLMs.

**Additional Comments On Reviewer Discussion:**

This is an extremely borderline paper, with scores (5,5,5,6), leaning towards rejection.

Reviewers generally agree with the assessment that the paper doesn't provide any new fundamental learnings or important results. As quoted directly from reviewers:
  * Reviewer tYHA: "studies focus primarily on minor hyperparameter adjustments, leading to conclusions that align with conventional expectations"
  * Reviewer rZP2: "Also, the paper lacks a hypothesis-driven structure that ties the findings to the central research question, which appears more like a tech report rather than a research paper"
  * Reviewer PDVC: "This work can be considered as a kind of grid-search process, and doesn't proposed new techniques in terms of methodology."

The question then lies as to whether it's a significant contribution that the paper shows that regular RL methods do work on multimodal reasoning. Regarding this contribution, the reviewers raised the lack of additional base models and benchmarks. This was addressed most recently in the rebuttal phase, where the authors provided many more experiments on additional evals and models.

However, Reviewers still found the contribution not enough post-rebuttal, and decided to maintain their score. Following this, the recommendation is to reject.

---

### Decision · Program_Chairs · 2025-01-22

Reject